# The Potential of Flavonoids and Flavonoid Metabolites in the Treatment of Neurodegenerative Pathology in Disorders of Cognitive Decline

**DOI:** 10.3390/antiox12030663

**Published:** 2023-03-07

**Authors:** James Melrose

**Affiliations:** 1Raymond Purves Laboratory, Institute of Bone and Joint Research, Kolling Institute of Medical Research, Faculty of Health and Science, University of Sydney at Royal North Shore Hospital, St. Leonards, NSW 2065, Australia; 2Graduate School of Biomedical Engineering, University of NSW, Sydney, NSW 2052, Australia; 3Sydney Medical School, Northern Campus, University of Sydney at Royal North Shore Hospital, St. Leonards, NSW 2065, Australia

**Keywords:** therapeutic treatment of neurological disorders, gut-brainaxis, protocatechuic acid, urolithins, γ-valerolactones, autism, bipolar disorder, Alzheimer’s disease, Parkinson’s disease

## Abstract

Flavonoids are a biodiverse family of dietary compounds that have antioxidant, anti-inflammatory, antiviral, and antibacterial cell protective profiles. They have received considerable attention as potential therapeutic agents in biomedicine and have been widely used in traditional complimentary medicine for generations. Such complimentary medical herbal formulations are extremely complex mixtures of many pharmacologically active compounds that provide a therapeutic outcome through a network pharmacological effects of considerable complexity. Methods are emerging to determine the active components used in complimentary medicine and their therapeutic targets and to decipher the complexities of how network pharmacology provides such therapeutic effects. The gut microbiome has important roles to play in the generation of bioactive flavonoid metabolites retaining or exceeding the antioxidative and anti-inflammatory properties of the intact flavonoid and, in some cases, new antitumor and antineurodegenerative bioactivities. Certain food items have been identified with high prebiotic profiles suggesting that neutraceutical supplementation may be beneficially employed to preserve a healthy population of bacterial symbiont species and minimize the establishment of harmful pathogenic organisms. Gut health is an important consideration effecting the overall health and wellbeing of linked organ systems. Bioconversion of dietary flavonoid components in the gut generates therapeutic metabolites that can also be transported by the vagus nerve and systemic circulation to brain cell populations to exert a beneficial effect. This is particularly important in a number of neurological disorders (autism, bipolar disorder, AD, PD) characterized by effects on moods, resulting in depression and anxiety, impaired motor function, and long-term cognitive decline. Native flavonoids have many beneficial properties in the alleviation of inflammation in tissues, however, concerns have been raised that therapeutic levels of flavonoids may not be achieved, thus allowing them to display optimal therapeutic effects. Dietary manipulation and vagal stimulation have both yielded beneficial responses in the treatment of autism spectrum disorders, depression, and anxiety, establishing the vagal nerve as a route of communication in the gut-brain axis with established roles in disease intervention. While a number of native flavonoids are beneficial in the treatment of neurological disorders and are known to penetrate the blood–brain barrier, microbiome-generated flavonoid metabolites (e.g., protocatechuic acid, urolithins, γ-valerolactones), which retain the antioxidant and anti-inflammatory potency of the native flavonoid in addition to bioactive properties that promote mitochondrial health and cerebrovascular microcapillary function, should also be considered as potential biotherapeutic agents. Studies are warranted to experimentally examine the efficacy of flavonoid metabolites directly, as they emerge as novel therapeutic options.

## 1. Introduction

The vagus nerve is the longest and most complex of the 12 cranial nerves in the human body and a major component of the parasympathetic nervous system. It provides autonomic control and functional regulation of internal organs, controlling such fundamental processes as digestion, the pulsatile behaviour of the heart (which controls blood circulation), and the behavior of the muscle systems that control the respiratory system [1,2,3]. The autonomic nervous system also controls reflex actions, such as coughing, sneezing, and swallowing, and coordinates with the sympathetic nervous system to achieve organ homeostasis. The vagus nerve has important roles as a line of communication between the gut microbiome and linked organ systems [4]. In infancy, the microbiome contributes to the education of the immune system by exposing it to a range of epitopes, leading to a diverse recognition system that can identify self- from non-self-preventing sensitivities to food epitopes in adulthood, and also to the development of auto-immune disorders and life-threatening allergies [5]. The microbiota, gut, and brain communicate through the vagus nerve In a bidirectional communication system in what has been termed the gut-brain axis (Figure 1) [5]. The vagus nerve is a mixed nerve containing 80% afferent and 20% efferent fibres that deliver important instructive information in the form of vesicular neurotransmitters using a sophisticated transport system [6]. Activated nerves transport neurotransmitters to the synaptic gap where neurotransmitters release transducer signals and motor functions mediated by neural networks. Bioactive compounds are transported to the brain by efferent vagal fibres to stimulate specific brain regions. Vagal stimulation has been used to treat neuropsychiatric disorders [7].

Although controversial, psychedelic drugs belong to a general class of compounds known as psychoplastogens, which robustly promote structural and functional neural plasticity in key neural circuits that in practice have been shown to be beneficial to brain health [8]. Progress in this branch of medicine has historically been hampered by legislation banning the use of such psychotropic medications [9]. Psychedelics are serotonin 2A receptor agonists that can lead to profound changes in perception, cognition, and mood, and display a potential in the treatment of mental health brain disorders that is unlike any other treatments currently available [10]. Psychedelics can produce sustained therapeutic benefit following a single administration, and also have broad therapeutic value and efficacy in the treatment of disorders, such as depression, post-traumatic stress, anxiety disorders, and addictive substance and alcohol abuse disorders [11]. A number of flavonoids have been identified with an ability to regulate neural functional properties of potential therapeutic value in the treatment of neurological disorders [10,12,13,14,15,16,17,18,19,20,21,22,23,24,25,26,27,28,29,30,31,32,33,34,35]. One class of flavonoid metabolite (urolithins) shows particular promise in the treatment of neurodegenerative disorders and in the provision of general health and wellbeing [20,36,37,38]. Elligatannins are degraded to ellagic acid, which is further processed to the urolithins by gut bacteria. Human intestinal bacteria capable of producing isourolithin A from ellagic acid have been isolated [33].

Traditional complimentary medical herbal infusions have been used for centuries to treat pain symptoms in the treatment of headaches [39,40], as antipyretics in the treatment of fevers [41], and have also been shown to be beneficial in the treatment of neurological symptoms in disorders of functional cognitive decline [42,43,44]. In Chinese traditional medicine, the liver is considered to be of central importance in the regulation of the Qi vital life force, which in therapeutic procedures is re-directed through the meridians to organ systems to re-balance vital life forces [45,46] Chinese medicinal herbal preparations are considered to re-balance the harmony of the opposing life elements of the yin and yang [47,48]. The Qi represents the functional activities of the body classified as yin, while the vital control of these bodily functions is provided by the yang component [49]. The gut-liver-brain axis thus has a central role in Chinese medicinal doctrine and the benefits provided by Chinese herbal formulations. While there are no equivalent or plausible explanations of this abstract theory in Western medicine [50], the existence of a gut-liver, gut-lung, and gut-brain regulatory connection that exerts some measure of control over linked organ systems has received considerable attention [51,52,53,54,55,56,57,58]. Many studies have proposed gut-liver, gut-lung, and gut brain axes as potential routes of intervention in disease resolution. The vagus nerve provides communication between the brain and gut, facilitating cross-talk between the brain and gut microbiota, and is a major parasympathetic heart regulatory nerve. In the intestines, the vagus nerve regulates the contraction of smooth muscles and glandular secretions. The vagus nerve thus oversees crucial bodily functions, such as mood control, immune response, digestion, and heart rate [59,60].

## 2. Therapeutic Vagus Nerve Stimulation

The microbiota, gut, and brain bidirectionally communicate via the microbiota-gut-brain axis [5]. The vagus nerve transports microbiota metabolites through its efferent fibres to the CNS where a number of responses in neuronal cell populations occur. A cholinergic anti-inflammatory pathway in the vagus efferants dampens peripheral inflammation; flavonoid metabolites have antioxidant and anti-inflammatory properties and are bioavailable to neural cells thus positive outcomes can also be expected on brain cell populations. The vagus nerve of the parasympathetic nervous system oversees a vast array of crucial bodily functions, including control of mood, immune response, digestion, and heart rate [4]. It establishes a crucial connection between the brain and the gastrointestinal tract and sends information about the state of the inner organs to the brain, which aids in homeostasis of bodily functions. Vagus nerve stimulation is a promising supportive treatment for refractory depression, posttraumatic stress disorder, and inflammatory bowel disease, inhibits cytokine production, and positively effects beneficial monoaminergic brain signaling in psychiatric conditions, such as mood and anxiety [61,62,63], and also for the treatment of traumatically injured brain tissues [64,65,66,67,68,69]. Gut bacteria have beneficial effects on mood and anxiety through the bioactive factors they produce, which are transported by the vagus nerve to the brain. The transfer of information between the gut and the brain via the vagus nerve is a two-way communicative highway with afferent vagal fibers actually outnumbering efferent fibres to a significant degree [70,71]. Vagus nerve stimulation has been used to treat epilepsy [6,72,73] and depression [6,72,73,74,75], and to improve learning and memory [76]. Thus positive functional outcomes are achievable when the vagus nerve is used as a conduit to stimulate brain tissue [77].

Preclinical evidence firmly establishes bidirectional communication between the brain, gut, and the gut microbiome through at least three nerve communication channels [78]. The vagus nerve has a cholinergic anti-inflammatory pathway that dampens peripheral inflammation, decreases intestinal permeability, and may also modulate the microbiota cell populations [5]. A large number of studies highlight potential roles for microbial dysbiosis as a contributing factor in many chronic disorders [79]. The gut microbiota and brain communicate through the gut-brain axis [80], when disturbed this may contribute to the pathophysiology of neurodegenerative disorders [81,82,83]. Methylation of ingested dietary flavonoids increases their lipophilic character, facilitating transport by the vagal cholinergic pathway from the gut to the brain. Flavonoids have anti-inflammatory and antioxidant properties that inhibit neuroinflammation and improve brain health. Microbiome dysbiosis, including a low abundance of *Faecalibacterium* and *Bacteroides* sp. and decreased production of butyrate in the gut, may foster inflammation and may contribute to the underlying pathophysiology of bipolar disorder [84]. A disturbance in the autonomic nervous system may provoke and maintain gastrointestinal dysbiosis in autism spectrum disorder [85]. Emerging data has identified a link between gut microbiota dysbiosis and neurodegenerative disorders, such as PD, AD, and ALS [86]. Neuroinflammation is, therefore, now being increasingly recognized as a driver of neurodegenerative disease pathology [87,88]. Gut bacteria also have crucial roles to play in the maintenance and regulation of the immune system, thus alterations in gut microbial cell populations may detrimentally affect neuro-immune interactions, synaptic plasticity, and regulation of skeletal muscle activity. This opens up the possibility of translational interventional studies in the treatment of neurodegenerative disorders and the emergence of psychobiotic programmes [89,90,91,92].

## 3. Transporter Proteins in the Afferent Fibres of the Vagus Nerve

Gastrointestinal vagal afferent fibres outnumber efferent fibres in the vagus nerve, however, these convey sensory signals from the gastrointestinal tract to the brain. Numerous subtypes of gastrointestinal vagal afferents have been identified [93]. Stimulation of the vagus nerve has been used in the treatment of epilepsy and seizures, but also shows therapeutic potential in a range of other serious neurodegenerative disorders [4] and has found application in the treatment of inflammation [94], and also to combat the cytokine storm of ARDS in the COVID-19 disease [95]. Vagus nerve stimulation limits cytokine production and dampens systemic inflammation and inflammation-induced lung tissue damage [1]. Neurotransmitters are synthesized in the cytoplasm of nerves and by the gut microbiota and are transported in secretory vesicles in nerves for regulated release at synaptic membrane interfaces with communicating neural networks (Figure 2) [96]. Neurotransmission depends on the efficient regulation of the transport and release of chemical transmitter molecules. Neurotransmitters are packaged into specialized secretory vesicles in neurons and neuroendocrine cells, and these are transported by specific vesicular transporter proteins [97]. The vagus nerve contains transporter proteins that send amino acids and sugar nutrients to the brain generating satiety responses to hunger [98] and signals that regulate hunger responses/food intake and the production of gastric and pancreatic secretions [99]. Relatively little is published, however, on transport systems for flavonoid or flavonoid metabolites generated in the gut to the brain. Most of the flavonoid metabolites generated by the gut microbiota are of a similar size and chemical composition to neurotransmitter compounds and nutrient derived components, and thus may also be shuttled by these transporter systems in the efferent fibres of the vagus nerve to the brain (Figure 3). In vitro experiments show that many of the flavonoid and flavonoid metabolites have antioxidant and anti-inflammatory effects on neurons, stimulate the biogenesis of mitochondrial components, and have vasodilatory properties beneficial to the brain microvasculature. Transport of these gut components to the brain may thus be the therapeutic basis of vagal stimulation and its beneficial properties in the treatment of neurodegenerative conditions, and a link between the diet and autism, bipolar and other neurological disorders [4,51,57,59,80,81,83,84,88,92,100].

Neural transmitters are small bioactive molecules that are carried in synaptic vesicles in nerves. When a nerve is activated (Figure 2) a number of proteins transport and release neurotransmitters at the synaptic gap. Figure 3 shows some bioactive flavonoid metabolites that we propose are transported by nerves and have stimulatory effects when delivered to neural cell populations in the brain.

## 4. Neuroregulatory Properties of Flavonoids

Figure 2 demonstrates the crucial role of Ca^2+^ entry into neurons in their activation and transport of neurotransmitters that transduce signals in neural networks. Phenolic compounds (numbering in excess of 8000 compounds) have long been known to have medicinal properties. In this review, we concentrated on a sub-category of the phenolics flavonoids), which have been categorized into six sub-categories (Figure 4a,b). These have a generic 3 ring structure, as shown in Figure 4c.

Some flavones and flavone metabolites have been observed to modulate neural processes. Catechin and procyanidin flavan-3-ols are transported by nerves and effect adipose tissue mediated nerve activation [101]. The flavonoid isoliquiritigenin activates GABA_B_ receptors, decreasing entry of Ca^2+^ into rat cerebrocortical nerves through voltage-gated Ca^2+^ channels, affecting glutamate transport and release from synaptic vesicles, and the transduction of neurotransmitters in neural networks [102]. Genistein isoflavone, a tyrosine kinase inhibitor, reduces Ca^2+^ influx through T-type Ca_V_3.3 voltage-gated ion channels affecting nerve activation [103]. Dysfunction of T-type calcium channels is associated with epilepsy, neuropathic pain, cardiac problems, and major depressive disorders. Molecular agents that modulate the T channel function may thus be therapeutic. Baicalin ameliorates neuropathic pain by suppressing TRPV1 up-regulation and ERK phosphorylation in DRGs [104] and modulates the dopamine system, thus modulating behavior seen in attention deficit hyperactivity disorder [105]. The pharmacological properties of baicalin are associated with the synthesis, vesicular localization, transport, and release of dopamine from synaptic vesicles. Naringenin has antinociceptive analgesic effects through its ability to inhibit NaV1.8 voltage-gated sodium channels preventing nerve activation and the generation of neuropathic pain signals [106]. Green tea EGCG has vasodilatory effects, reduces blood pressure, and activates zebrafish TRPA1 channels in sensory neurons triggering CGRP release, a potent vasodilator [107]. Diabetic peripheral neuropathy and neuropathic pain are major public health issues impacting on quality of life. TRPV1 has a crucial role in nociceptive transmission of pathological pain. Baicalin is an antioxidant flavonoid whose analgesic effects on spinal neuropathic pain are apparently mediated through TRPV1 [108]. The excessive release of glutamate critically effects the neuropathology of acute and chronic brain disorders. Apigenin reduces presynaptic Ca^2+^ entry mediated by the Cav2.2 (N-type) and Cav2.1 (P/Q-type) channels, thereby inhibiting glutamate release from the rat hippocampal nerve terminals [109]. Myricetin, a natural neuroprotective flavonoid, also inhibits the release of glutamate from nerve terminals (synaptosomes) of the rat cerebral cortex through effects on Cav2.2 (N-type) and Cav2.1 (P/Q-type) channels by attenuating voltage-dependent Ca^2+^ entry and activation of nerves that generate pain responses [110]. Kaempferol-3-rhamnoside and rosmarinic acid also inhibit synaptic glutamate release, inhibiting nerve activation and generation of pain responses [111,112].

Neuroinflammation has detrimental effects on neurons and contributes to the pathology of neurodegeneration. The beneficial antioxidant properties of flavonoids [113,114] is attributable to their ability to inhibit lipoxygenase (LOX), cyclooxygenase (COX), myeloperoxidase (MPO), NADPH oxidase, and xanthine oxidase (XO). Flavonoids also stimulate free radical scavenging enzymes, such as superoxide dismutase (SOD) and catalase (CAT), which reduce levels of free radical oxygen species (ROS), including superoxide radical, hydroxyl radical, and singlet oxygen. This involves conversion of the superoxide ion into hydrogen peroxide by SOD, and this is converted into water and oxygen by CAT. The nuclear factor erythroid 2-related factor 2 (Nrf2)/antioxidant response element (ARE) pathway is an important cell signaling pathway responsible for the maintenance of redox homeostasis in humans [115,116]. Nrf2 is a master regulatory pleiotropic transcription factor that controls hundreds of genes in the phase II antioxidant response, controlling a multitude of cytoprotective genes responsive to oxidative stress and inflammation. Activation of Nrf2 produces antioxidant, anti-inflammatory, and neuroprotective effects, and is a critical component in the regulation of oxidative stress and anti-inflammatory responses in the CNS [117]. Luteolin, apigenin, quercetin, myricetin, rutin, naringenin, epicatechin, and genistein are all capable of activating the Nrf2/ARE pathway contributing to neuroprotection and the homeostasis of the CNS [118]. Table 1 illustrates further examples of flavonoids that also induce Nrf2 expression and its protective effects.

Quercetin occurs in plants as a glycosylated compound called rutin (Figure 5a), however, when ingested, rutin is converted to the aglycone form (Figure 5b) and modified by glucuronidation (Figure 5c) or sulfation (Figure 5d).

## 5. Natural Flavonoids Used in the Treatment of Neurodegenerative Conditions

Traditional Chinese medicine using herbal preparations have been used for centuries in complementary alternative medical practices [155,156]. Traditional Chinese herbal preparations are extremely complex mixtures of pharmacological agents often derived from up to seven different herbs. With the modern analytical techniques now available, attempts have been made to demystify these preparations to identify specific compounds and their mechanisms of action and to better understand their operational pharmacologic networks. The aim is to put these traditional medical practices on a more scientific basis to determine if they can be applied in Western medical practices. Network pharmacology, molecular docking, and in vitro cell-based investigations have identified a number of active components in these herbal preparations that could potentially provide a therapeutic effect [157,158,159,160,161].

## 6. Traditional Chinese Medicinal Formulations Used to Treat Alzheimer’s Disease

### 6.1. LeZhe

The *Menispermaceae* are small woody flowering climbing shrubs that contain a wide range of pharmacologically active benzylisoquinoline alkaloids, lignans, flavones, flavonols and pro-anthocyanidins. *Tinospora sinensis* is a member of the *Menispermaceae* family used in traditional Chinese medicine to treat AD. The formulation used, *LeZhe*, is a nerve calmative detoxifying antipyretic. Network pharmacology and molecular docking studies have identified *LeZhe’s* active compounds and molecular targets. Screening of DrugBank, Therapeutic Target Database and published AD studies have been used to identify pharmacological agents of interest. Kyoto Encyclopedia of Genes and Genomes (KEGG) target pathway enrichment analyses using Database for Annotation, and Visualization and Integrated Discovery (DAVID) have been undertaken and the neuroprotective properties of *T. sinensis* bioactive compounds have been evaluated in PC12 primary hippocampal neural cultures where injury had been induced using Aβ_25-35_. A total of 105 *T. sinensis* compounds and 38 molecular target proteins were identified. The main bioactive compounds of *LeZhe* include alkaloids such as berberine, a tetracyclic isoquinoline alkaloid derived from tyrosine, the aromatic amide aurantiomide, a quinazoline alkaloid; coumaroyl tyramine, hydroxycinnamic acid; trans-syringin, a β-D-glucoside derivative; and 3-dimethyl phillyrin phenylpropanoids. Phillyrin is a lignan produced by the endophytic fungus *Paraconiothyrium* sp. associated with the Chinese medicinal plant *Forsythia suspensa* with reported anti-pyretic detoxifier, antioxidant, anti-infective, anti-inflammatory, and antiviral properties [162] (Figure 6a–f). Many of these compounds can penetrate the blood brain barrier. Molecular targets of *T*. *sinensis* herbal compounds include Protein kinase B (AKT), Phosphoinositide 3-kinase (PI3K), Tyrosine-protein kinase JAK1 (JAK1), mammalian target of rapamycin (mTOR), TNF-α, Neuronal NOS, and the cholinergic function-related proteins, α4-Nicotinic acetylcholine receptor (α4 nAChR) and Muscarinic acetylcholine receptor M1 (Muscarinic M1). Inflammation and cholinergic dysfunction are targeted through PI3K/Akt, neurotrophic factor (NTF), Hypoxia-inducible factor 1 (HIF-1), mTOR, TNF and insulin resistance (IR) signalling pathways [160]. Significant improvement in PC12 cell survival and inhibition of apoptosis of Aβ_25-35_ injured primary hippocampal neuron cell cultures demonstrates the therapeutic potential of *T. sinensis* preparations in AD through a complex multi-compound-multi-target regulatory network however details still need to be unraveled of the mode of action of specific bioactive compounds [161]. Several bioactive flavonoid components have been identified in LeZhe preparations (Figure 6a–f) Berberine has anti-diabetic, anti-inflammatory properties, lowers blood sugar levels, causes weight loss and lowers blood pressure [163]. Berberine protects against TNF α induced inflammation in adipocytes [164] and is neuroprotective suppressing NF-κB-mediated neuroinflammation and pyroptosis [165]. Aurantiomides A-C isolated from the sponge-derived fungus *Penicillium aurantiogriseum* await detailed characterisation [166]. Syringin is a natural anti-inflammatory glucoside that attenuates NO production in LPS-stimulated RAW264.7 cells and has anti-oxidant and anti-cancer properties [167]. Phillyrin is a heterocyclic lignan glycoside flavonoid that attenuates TNF α-mediated insulin resistance and accelerated lipolysis by adipocytes [168,169].

### 6.2. Shuang-Huang-Lian Herbal Preparations

Shuang-Huang-Lian is listed in the Chinese pharmacopeia for the treatment of respiratory infections and has purported antiviral SARS-CoV-2 activity [170]. Baicalin and its aglycone form, baicalein, are two ingredients of Shuang-Huang-Lian herbal preparations (Figure 6g,h), and have been identified as BBB penetrating noncovalent, nonpeptidomimetic inhibitors of SARS-CoV-2. 3CLpro and may also be beneficial in the treatment of attention deficit hyperactivity disorder. Baicilin and baicalein are positive allosteric modulators of the benzodiazepine/non-benzodiazepine sites of the GABA_A_ receptor, [171,172] providing anxiolytic [173,174,175] and anticonvulsant properties [176,177,178] and are neuroprotective prolyl endopeptidase inhibitors [179]. Prolyl endopeptidase/oligopeptidase (PEO) is implicated in a number of neurological disorders of the CNS, such as amnesia and stages of depression, and has roles in lithium sensitive signal transduction and depression [180,181]. PEO is implicated in neurodegeneration and neuroinflammation, and is considered a drug target for the enhancement of memory in dementia [182]. Inhibition of PEO reduces α-synuclein aggregation in PD [183]. increases α-synuclein degradation by neural cells [184]. and reduces α-synuclein toxicity [181]. In silico approaches inspired by the natural flavonoid baicilin, baicalein, and wogonin PEO inhibitors, are being used to produce synthetic PEO inhibitors of improved efficacy to reduce α-synuclein expression [185] (Figure 6i). A deficiency of PEO in mice reduces anxiety-like behavior and improves cognitive function [186], thus this approach is likely to be successful in the treatment of human neurological disorders. Such PEO inhibitors are of a small molecular weight similar to that of the neurotransmitters that are known to be transported by the vesicular transport system of nerves and are also expected to be transported by a similar mechanism.

### 6.3. Chaihu-Shugan-San

Chaihu-Shugan-San (CSS) is another well-known herbal antidepressant Chinese medicine that may also be beneficial in the treatment of cognitive dysfunction in AD [159]. Active compounds in CSS have been screened using the Traditional Chinese Medicine Systems Pharmacology database. Compound-related targets retrieved using the SwissTarget Prediction database identified major depressive disorder (MDD)-related targets using the DisGeNET Therapeutic Target and DrugBank databases. The identification of the active compounds in CSS affecting MDD targets has permitted the construction of a MDD target network in chronic unpredictable mild stress (CUMS) mice. Molecular docking established the binding affinities of these bioactive CSS compounds. Multi-target mechanisms of action of CSS compounds in network pharmacology identified a total of 152 active compounds, 520 predicted biological and 160 AD-specific targets. Sixty key targets providing beneficial effect in AD treatment were nuclear or cytoplasmic proteins with regulatory roles in PI3K-Akt, MAPK, and HIF signaling pathways in GO function and KEGG pathway enrichment analysis. Pre-treatment of PC12 neural cell cultures with CSS reduced Aβ-induced neural cell death and apoptosis. Increased phosphorylation of Akt and decreased pGSK3β/GSK3β levels in the hippocampus of CUMS mice established effects on PI3K/Akt signalling, and improved depressive-like behavior and neurogenesis of CSS in CUMS mice. Flavonoids identified in CSS include quercetin, luteolin and kaempferol; these warrant further examination in the treatment of AD.

### 6.4. Qingfei Paidu and Ma Xing Shi Gan

Qingfei Paidu and Ma Xing Shi Gan antiviral decoctions used to treat COVID-19 and AD in traditional Chinese medicine are also of considerable complexity. Molecular networking of mass spectrometry data has identified a number of bio-active flavone and chalcones present in these formulations [187] (Figure 7). Hesperidin, glycyrrhizic acid, baicalin, baicalein, naringin, phillyrin, quercetin, luteolin, kaempferol, licochalcone B, and mangiferin have all been identified in these formulations. Further studies are required to fully decipher all therapeutic bioactive component combinations and their interactions in the pharmacological networks. This initial study, however, made significant inroads into better understanding the complex therapeutic basis of these traditional Chinese herbal medications, but further work is required to fully understand how these components provide their therapeutic effect.

## 7. Complex Heterocyclic Polyphenolic Precursor Compounds That Are Processed by the Gut Microbiome Releasing Bioactive Metabolites

### 7.1. Eligatannins

Ellagitannins (ETs) are polyphenol compounds that are abundant in some fruits (blackberries, raspberries, strawberries), nuts (walnuts and almonds), and pomegranatesm, and have been used in complimentary medicine for centuries. ETs represent one of the most diverse groups of plant phenolics encompassing over 1000 natural bioactive compounds [188,189]. The gut microbiome converts ETs to ellagic acid (EA). EA has a variety of health benefits related to the protection it provides from oxidative stress [20,188,190,191,192]. EA is reported to have a low water solubility and bioavailability, however, when it is converted to urolithin A (UA) by the gut microbiome, UA retains the biological activities of EA and has high solubility and bioavailability (Figure 8). Urolithins are biologically active compounds exhibiting strong antioxidant effects [193,194,195,196] and anti-inflammatory [196,197] and neuroprotective properties [100,198]. Punicalagin, chebulinic acid, and chebulagic acid are complex polyphenolic ellagitannins that occur in pomegranate and are degraded to form EA by the gut microbiome (Figure 8). EA is further degraded to the urolithins; these are not synthesised or generated by mammalian cells and have antioxidant and anti-inflammatory properties [20,100,190,191,192,193,194,195,196,197,198]. Panduratin A is an antioxidant polycyclic chalcone phenolic that has also been identified in pomegranate and in the Thai medicinal plant *Boesenbergia rotunda* that has reported antiviral properties against SARS-CoV-2 [199].

EA is a candidate drug for the treatment of traumatic brain injury and neurodegenerative disorders, due to its neuroprotective properties mediated by inhibition of the PI3K/Akt/mTOR and Akt/IKK/NFκB signaling pathways, reducing neuroinflammation and enhancing autophagy [20,200,201,202,203]. The urolithins may be the bioactive metabolites that provide these beneficial therapeutic properties for EA [100,204,205,206]. EA can modulate the expression of the proinflammatory cytokines IL-1β,TNF-α, and IL-17 [20,193,196,197,206]. EA down regulates IL and lipid peroxidation, improves cognitive functions, and is provides neuroprotective benefits by scavenging free radicals and regulating antioxidant enzymes [100,198]. Urolithin species have neuroprotective properties through their antioxidant properties and ability to inhibit Aβ_25-35_-induced neurotoxicity and monoamine oxidase [20,201,207,208].

### 7.2. The Urolithins

Urolithin A is a benzocoumarin metabolite produced by the gut microbiome by digestion of ellagic acid and ellagitannins found in dietary pomegranates, strawberries, raspberries, and walnuts. Urolithin A does not occur freely in dietary foods, nor is it produced by mammalian enzyme systems [33,195]. Urolithin A is a natural prebiotic that promotes mitophagy, mitochondrial biogenesis, and metabolic function, impacting on muscle health in preclinical models of aging and in the elderly and middle-aged. Urolithin A improves mitochondrial function in the articular chondrocytes of diarthrodial joints, reducing disease progression in a mouse OA model, and inhibits cartilage degeneration, synovial inflammation, and the pain associated with this condition [36].

### 7.3. Hydroxybenzoic Acids

Protocatechuic acid has potent anti-inflammatory properties [209] and activates the master transcription factor, nuclear factor erythroid 2-related factor 2 (Nrf2), [210] through Jun kinase (JNK) modification of the Nrf2 signalling system [210]. Nrf2 binds to antioxidant response elements in the promoter regions of a large number of genes encoding cytoprotective proteins [211]. Activation of Nrf2 results in the induction of a large range of phase II detoxifying antioxidant enzyme systems [212] and also inhibits the NLR family pyrin domain containing 3 (NLRP3) inflammasome [213]. NLRP3 operates as part of the innate immune response as a pattern recognition receptor recognizing pathogen associated molecular patterns (PAMPs). Inflammasomes are multiprotein complexes in the innate immune system that induce inflammation in response to pathogenic organisms and stress. Activation of proinflammatory caspases, such as caspase-1, leads to an upregulation in proinflammatory cytokine levels, such as IL-1, -18, and -33, which promote neuroinflammation and pathological changes in brain tissues [214]. The NLRP3 inflammasome has important roles in the pathology of neurodegeneration and is a logical therapeutic target to alleviate the damaging aspects of neuroinflammation. Protocatechuic acid has significant potential in the inhibition of the NLRP3 inflammasome. Urolithin A is reported to improve mitochondrial and neuronal cell health [36,200,215,216,217].

## 8. Vasodilatory Flavonoids

Flavonoids exert positive beneficial effects on the cardiovascular system through their vasodilatory properties and ability to regulate apoptotic processes in the endothelium [218]. Hesperidin has been used for decades to treat vascular insufficiency in tissues [219].

The potential use of flavonoids and flavonoid metabolites to improve cerebrovascular circulation could prove to be useful to improve the treatment of neurodegenerative conditions.

Quercetin displays useful cardiovascular properties, however, its low bioavailability may limit its therapeutic application. The bioavailability of quercetin in the systemic circulation is low, with maximum plasma concentrations rarely exceeding 1 μM after consumption of 80–100 mg quercetin equivalents [220,221]. However when non-absorbed quercetin reaches the colon, it is subjected to further processing by the gut microbiome [218]. This includes C-ring cleavage, dihydroxylation, and decarboxylation, generating quercetin metabolites, such as 3,3-dihydroxyphenyl propionic acid and 3,4-dihydroxyphenyl acetic acid, which display vasodilatory properties in animal models and decrease arterial blood pressure [222,223].

EGCG catechins and epicatechins have beneficial effects on vascular function [224], cardioprotective effects through the reduction of systolic and diastolic blood pressure, and positive effects on the cerebrovascular circulation, which improves therapeutic treatment of neurodegenerative disorders [225,226]. Studies have also shown that flavonoid metabolites can have different biological and antioxidant properties and efficacy than the parent flavonoid. Modifications of the ingested flavonoid by methylation, glucuronidation, or sulphation also influences the biological activity of the flavonoid and has significant effects on their antioxidant and anti-inflammatory properties, and how they affect expression of cell-adhesive proteins [227]. NF-κB activation is the main transcription factor mediating TNFα-induced expression of inflammatory genes [228]. Pharmacological inhibitors of NF-κB activity, however, may also act through the stimulation of the Nrf2 pathway [229]. (-)-Epicatechin (EC) is metabolized by microbiota in the large intestine producing a major metabolite 5-(3′,4′-dihydroxyphenyl)-γ-valerolactone (3,4-diHPV). EC and 3,4-diHPV both activate Nrf2-mediated gene expression, however, 3,4-diHPV shows higher potency in the upregulation of Nrf2 gene expression than EC. Conversion of EC to 3,4-diHPV by the gut microbiota improves the overall health-promoting effects of EC consumption due to this ability to selectively promote Nrf2 pathway activation [230].

### The Anthocyanidins

Anthocyanins are antioxidant plant flavonoids with reported beneficial health-promoting effects in a number of chronic diseases [231]. Studies investigating anthocyanin absorption by Caco-2 intestinal cells report very low absorption of these compounds. The gut microbiome, however, converts the anthocyanins to protocatechuic acid and phlorglucinaldehyde, and these may be the pharmacologic bioforms that exert the purported therapeutic effects of the anthocyanins [232] (Figure 9).

Alzheimer’s disease (AD) is a serious and progressive neurodegenerative disorder of the elderly. Genetic, environmental, and lifestyle factors are associated with the pathogenesis of AD, leading to deleterious effects on the brain’s neuronal cell population manifested as cognitive dysfunctions, behavioural disability, and psychological impairment. Accumulation of amyloid beta (Aβ) peptides and neurofibrillary tangles in AD-affected brains are hallmarks of this disease. Several reports indicate flavonoids improve cognitive functions, inhibit or delay the formation of pathological amyloid beta aggregates and neurofibrillary tangles, thus improving neural function.

Current research has uncovered that dietary use of flavonoid-rich food sources essentially increases intellectual abilities and postpones or hinders the senescence cycle and related neurodegenerative problems, including AD [233]. During AD pathogenesis, multiple signalling pathways are involved, and to target a single pathway may relieve the symptoms but not provide a permanent cure [233,234]. Flavonoids scavenge free radical species (ROS), however, upon reaction with ROS, the antioxidant capacity of flavonoids can become compromised. Recent evidence for at least some flavonoids shows that the oxidation of reactive phenolic residues can in fact enhance their antioxidant properties. This antioxidant activity arises from generation of metabolites that activate the Nrf2-Keap1 pathway [233,234], upregulating the cell’s endogenous antioxidant capacity, by the prevention of activation of prooxidant and proinflammatory NF-κB pathways [235]. Flavonoid metabolites, such as protocatechuic acid [236,237] and urolithin A [33,36,195] generated by the gut microbiome, also have potent direct antioxidant activities or provide mitochondrial protection by promoting mitochondrial biogenesis and metabolic activity, enhancing neural cell activity in the CNS in neurodegenerative conditions [203,238].

## 9. Natural Flavonoids and Multifunctional Analog Derivatives Used in Western Medicine to Treat Neurodegenerative Conditions

### 9.1. Hesperidin/Hesperitin

Hesperidin’s antioxidant, anti-inflammatory, and neuroprotective properties are useful in the treatment of neurodegenerative conditions [239] and have inspired the development of therapeutic multifunctional flavone and chalcone analogs of improved efficacy. Hesperitin also has considerable potential in the treatment of neurological disorders and has inspired the development of multifunctional agents of improved efficacy [26,233,234]. The central position of chalcones in medicinal chemistry, and its amenability to chemical modification, facilitates its use as a template for the development of multifunctional analog chalcone/flavone forms of improved efficacy in a number of depressive neurodegenerative disorders, including PD and AD. A multi-tier flavone screening protocol employing molecular docking for BACE1 inhibitory, and antiamyloidogenic and antioxidant activities, demonstrates hesperidin as a multi-potent phytochemical in AD therapeutics [240,241].

Hesperidin is a high affinity BACE1 inhibitor providing complete inhibition of amyloid fibril formation, moderate ABTS(+) radical scavenging, and strong hydroxyl radical scavenging activity [240]. Inhibition of BACE1 and Aβ aggregation occurs by binding close to the catalytic aspartate dyad-constraining BACE1, precluding APP recognition and inhibiting amyloid fibril formation, Aβ_25-35_ induced ROS generation, and mitochondrial dysfunction [242]. Mitochondrial dysfunction and oxidative stress both induce pathological neurodegenerative changes contributing to the development of AD [242]. Hesperidin inhibits Aβ-induced cognitive dysfunction, oxidative damage, and mitochondrial dysfunction in mice, reduces learning and memory deficits, and improves locomotor activity. Increased phosphorylation of GSK-3β by hesperidin, reduced mitochondrial dysfunction, and increased antioxidative defence improve cognitive function in the APPswe/PS1dE9 transgenic mouse model of AD [242]. Hesperidin also inhibits the development of neurodegenerative disease by elevating expression of neural growth factors and endogenous antioxidant defence, reducing the impact of neuroinflammatory and apoptotic pathways. A limited number of human clinical trials have shown that hesperidin-enriched dietary supplements significantly improved cerebral blood flow, cognition, and memory performance [243]. Cerebral ischaemic injury and degenerative pathology in AD are linked, hesperidin downregulates Bcl-2 and Akt/PI3K, protecting against Aβ_25-35_-induced apoptotic neurotoxic effects [243]. Oxidative stress and inflammation have pivotal roles in the pathophysiology of AD and are attenuated by hesperidin in APP/PS1 mice, resulting in a reduction in ROS, LPO, and increased activity of HO-1, SOD, catalase, and GSH. This inhibits neuroinflammation by decreasing TNF-α, C-reactive protein, MCP-1 levels, and NF-κB activity [244]. Phosphorylation of Akt and GSK-3β are decreased by hesperidin and RAGE expression is inhibited, while the enhanced phosphorylation of IκBα and the nuclear translocation of NF-κB/p65 in APP/PS1 mice evidences neuroprotective properties by suppressing neuroinflammation [245].

### 9.2. Kaempferol and Luteolin

Plant secondary metabolite inhibitors that target monoamine oxidases may be useful in the treatment of depressive neurodegenerative disorders such as PD and AD [246]. Kaempferol and luteolin are selective human MAO-A inhibitors [247,248].

## 10. The Potential of Flavonoids in Tissue Repair Processes in a Biodiverse Range of Diseases in Linked Organ Systems

Literally hundreds of in vitro studies have demonstrated the antioxidant and anti-inflammatory tissue protective properties of flavonoids. A number of flavonoids have also been shown to have neuroprotective and neuroregenerative properties. Preclinical studies in rodent, pig, and monkey models of AD, PD, HD, and ALS [34,249,250,251,252,253,254,255] have also demonstrated that flavonoids have properties that counter neuroinflammation, prevent the neurotoxic effects of pathological protein aggregates of amyloid and hyperphosphorylated tau protein, free radical generation from peroxidation of lipids, and that brain tissues are rich in phospholipids that are susceptible to oxygen radical release during neuroinflammation. Some flavonoids act as monoamine oxidase inhibitors, combat apoptosis, are neuroprotective, promote neurogenesis and memory, and reduce cognitive decline [256,257,258,259,260,261,262,263,264,265,266,267]. Examination of the clinical trials that have been conducted on flavonoids demonstrates their diverse areas of action and therapeutic potential.

Despite their low bioavailability, positive responses have nevertheless been observed with several flavonoids in many clinical trials indicating their therapeutic potential.

## 11. Flavonoid Clinical Trials

### 11.1. Hesperidin/Hesperitin Trials

Hesperidin has antioxidant tissue protective properties [268] and may improve cerebrovascular circulation, cognitive function, and the clinical manifestations associated with ocular disorders [269]. Hesperidin can improve vascular health and treat hypertension, improves cardiovascular function, has tissue protective properties in type 2 diabetes [270], and may be useful in the control of obesity, acute hemorrhoidal disease [271], muscle metabolism [272,273], and has skin antiaging properties [274].

Hesperidin, alone or in combination with other citrus flavonoids, such as diosmin, has been used to treat vascular defects, such as hemorrhoids, varicose veins, and poor circulation (venous stasis). Preclinical studies have also demonstrated its beneficial effects in the treatment of neurodegenerative disorders [275]. A review of preclinical trial data showed the beneficial neuropharmacological potential of hesperidin, including anticonvulsant, antidepressant, antioxidant, anti-inflammatory, memory, and locomotor enhancing activities [275].

### 11.2. Epigallocatechin Gallate Clinical Trials

A significant number of clinical trials have been conducted on the EGCG polyphenolic flavonoid component of green tea. An examination of a selection of these trials [265,276,277,278,279,280,281,282,283,284,285] amply demonstrate the diverse biological properties of EGCG and its potential therapeutic applications. EGCG can potentially improve cognition in children with Down syndrome [276]. EGCG prevents skin dermatitis in skin cancer patients receiving radiotherapy [277], and topical application of EGCG improves the treatment of vitilago [278]. EGCG improves surgical skin scarring reducing mast cell numbers, improving blood flow, angiogenesis and the elastin content of skin samples (ISRCTN70155584) [282]. In an international standard randomized controlled trial (registration number ISRCTN 18643079), EGCG improved scar repair, scar thickness, hydration, and elasticity [280]. EGCG acutely enhances muscle microvascular blood flow in healthy young adults [281]. Combination therapy of EGCG with hesperidin prevents obesity [282]. EGCG supplementation improves blood pressure, lipid profiles, plasma atherogenic index, and oxidative status in type 2 diabetes [283] when used in a controlled clinical trial on subjects receiving a high fat diet improved lipid profiles. EGCG has been described as a potent natural inhibitor of fatty acid synthase [284]. EGCG shows promise in an animal model of AD in the regulation of α-, β-, γ-secretase activity, inhibiting tau phosphorylation, has antioxidative, anti-inflammatory, antiapoptotic activity, and inhibits AChE activity, all contributing to EGCG’s neuroprotective properties [265]. A double-blind placebo-controlled phase I clinical trial of the cognitive effect of EGCG on Fragile X syndrome (TESXF; NCT01855971) also showed improved memory and cognition [285].

The gut microbiome generates metabolites from EGCG that improve cerebrovascular function and have therapeutic utility in the treatment of neurodegenerative disorders.

Epicatechin is known to improve cognitive functions, lowering the risk of AD or stroke, however, the biologically active molecular forms of epicatechin that are responsible are poorly understood. γ-Valerolactone metabolites of EGCG are biologically active and can simultaneously modulate the expression of protein-coding and non-coding genes to effect cellular regulation, effecting cell adhesion, cytoskeleton organization, focal adhesion, cell signaling pathways, regulation of endothelial cell permeability, and their interactions with immune cells [225]. Two major EGCG metabolites generated by the gut microbiome and detected in plasma are 5-(4′-hydroxyphenyl)-γ-valerolactone-3′-sulfate and 5-(4′-hydroxyphenyl)-γ-valerolactone-3′-O-glucuronide [286]. γ-valerolactones have high bioavailability and anti-inflammatory properties, decrease blood pressure [287], and improve cerebrovascular blood flow, improving cognitive impairment in neurological disorders [288]. Cerebrovascular dysfunction can accelerate brain atrophy with ageing, reduce cognitive capability, and lead to an increased risk of stroke and neurodegenerative diseases, such as AD and dementia. Flavonoids, including EGCG, have been shown in animal models [286,289,290] to maintain neurocognitive function in aging rats, decrease the risk of development of AD and stroke in humans, and exert beneficial effects on cerebrovascular blood flow in dementia [291,292,293]. 

### 11.3. Anthocyanin Clinical Trials

A randomised placebo-controlled trial of the effect of purified anthocyanins on cognition in individuals at increased risk for dementia has been undertaken [294]. A phase II clinical trial (ClinicalTrials.gov, NCT0341903) also assessed intervention strategies to prevent or delay the onset of dementia, and a further phase III trial of anthocyanins is also planned [295]. A trial has also examined the effect of dietary anthocyanins on endothelial function and arterial stiffness in individuals of excess body weight [296]. The consumption of anthocyanin is reported to improve memory in older adults with mild cognitive impairment [297,298]. The effects of anthocyanins on inflammatory and metabolic responses in a high-fat diet with cyanidin and delphinidin reported to exert beneficial effects in unhealthy diets [299]. Cyanadins were also reported to improve lipid profiles and lowered systemic inflammation in subjects with cardiovascular risk factors (ClinicalTrials.gov, NCT number: NCT04084847) [300]. The effects of anthocyanin supplementation on platelet function in subjects with dyslipidemia are shown to attenuate platelet function dyslipidemia [301]. The beneficial effects of berry anthocyanin consumption on cognitive performance, vascular function, and cardiometabolic risk markers uncovered in clinical trials has recently been reviewed [302]. A randomized controlled trial of consumption of tropical fruits rich in anthocyanins has also shown improvement in cognitive function, learning, memory, mental acuity, flexibility, and visual-motor skills in middle-aged women [303]. A clinical trial of anthocyanins has been shown to decrease concentrations of TNF-α in older adults with mild cognitive impairment (Australian New Zealand Clinical Trials Registry: ACTRN12618001184268) [304]. A cross-over, randomized, double-blind clinical trial (Australian New Zealand Clinical Trials Registry, identifier no. ACTRN12620000437965) showed anthocyanins attenuated vascular and inflammatory responses in overweight older adults [305].

### 11.4. Quercetin Clinical Trials

Quercetin exhibits many beneficial properties in cell and tissue protection in disease processes and optimal tissue function. A significant number of clinical trials have been undertaken examining the therapeutic efficacy of quercetin. These include potential roles in the treatment of cognitive function and cerebral blood flow [306] and modulation of the progression of AD [307] and CNS viral infection [308]. The efficacy of quercetin in muscle physiology has been evaluated [309] and its roles in the modulation of IGF-I and IGF-II levels following muscle damage [310]. Quercetin antioxidant effects have been examined in metabolic syndrome [311], and its efficacy in the treatment of blood pressure and endothelial dysfunction and regulation of lipid profiles and inflammatory biomarkers in metabolic syndrome [312,313]. A meta-analysis has been conducted on randomised controlled human trials assessing the impact of quercetin on systemic levels of inflammation [314]. Quercetin has been used to target IL-1β and suppress apoptosis in vascular endothelial cells in the treatment of atherosclerosis [315,316], in the treatment of cardiovascular disease [317], and in inflammatory processes effecting quality of life in post-myocardial infarction [318]. Quercetin has also been examined in inflammatory processes in polycystic ovary syndrome [319], combination therapies in the treatment of endometriosis [320], and in antiviral applications in COVID-19 [321]. Quercetin has been examined in combination therapy with green tea polyphenols in the treatment of prostate cancer [322], and the safety of quercetin supplementation assessed in the treatment of chronic obstructive pulmonary disease [323].

## 12. Bioactive Quercetin Metabolites

The gut microbiome members *Escherichia coli, Streptococcus lutetiensis, Lactobacillus acidophilus, Weissella confusa, Enterococcus gilvus, Clostridium perfringens* and *Bacteroides fragilis* have all been shown to degrade quercetin into a number of metabolites with *C. perfringens* and *B. fragilis* having the strongest degradative capability in vitro [324]. Peng et al. also demonstrated the presence of 3,4-dihydroxyphenylacetic acid production by *C. perfringens* and *B. fragilis* demonstrating their quercetin degradative capacity [325]. Fecal gut bacteria also degrade rutin as a substrate in-vitro releasing 3,4-dihydroxyphenylacetic acid as a metabolite [326].

Quercetin metabolites produced by *C perfringens* and *B fragilis* display a strong statistically significant inhibitory effect on HCT-116 human colorectal carcinoma cells. *Weissella confusa* produces quercetin metabolites with strong cytostatic tumor inhibitory activity over the growth of both A549 human lung carcinoma cells and HeLa cells comparable to or stronger than the tumor inhibitory activity displayed by intact quercetin but are more readily bio-available [327]. *Eubacterium ramulus* isolated from human feces is a strictly anaerobic bacterium of the gastrointestinal tract. *E. ramulus* cleaves the ring system of several flavonols and flavones giving rise to the corresponding hydroxyphenylacetic and hydroxyphenylpropionic acids, respectively, as well as acetate and butyrate. *E ramulus* generates 3,4-dihydroxyphenyl acetic acid from the biotransformation of quercetin in vitro and in vivo [328].

Neurodegeneration induced by the pesticide rotenone can be countered by quercetin in an animal model of PD [329,330], and in a transgenic model of AD [331], it reduced the neurotoxic effects of β-amyloidosis and decreased tauopathy in the hippocampus and amygdala, improving cognitive functional recovery. Quercetin is a multifunctional therapeutic in the treatment of neurodegenerative disorders [12,329,330,332]. Bioconversion of quercetin by gut bacteria generates bioactive quercetin metabolites, such as 3,4-dihydroxyphenylacetic acid and protocatechuic acid [333]. Protocatechuic acid is also a major metabolite of complex polyphenols, such as the anthocyanins and proanthocyanins [333,334,335,336]. Polyphenolic metabolites that also arise during flavonoid metabolism, such as 3,4-dihydroxyphenylacetic acid, can positively influence beneficial gut bacterial populations, such as *Bifidobacterium* spp., *Lactobacillus* spp. and *Bacteroides* spp., and inhibit colonization by the pathogenic bacteria *Fusobacterium varium*, *Bilophila*, and *Enterobacteriaceae*, thus promoting gut health [334] and enhancing the expression of several phase II drug-metabolizing enzymes that lower oxidative species in tissues.

## 13. Catechin Metabolites

Human phenyl-γ-valerolactone is a major metabolite of flavan-3-ols produced by gut bacteria. Phenyl-γ-valerolactone has neuroprotective properties and inhibits neurotoxic protein aggregate deposition, such as amyloid and tau in brain tissues [337], and promotes memory retention, preventing cognitive decline in an AD mouse model [338]. Phenyl-γ-valerolactone also improves endothelial cell function and cerebral blood flow [339,340,341]. Cerebral blood vessels are lined with endothelial cells and these form the blood–brain barrier (BBB). Endothelial dysfunction constitutes a crucial event in the pathophysiology of neurodegenerative disorders and cognitive impairment. Neuroinflammation can lead to neurodegeneration, endothelial cell dysfunction, defective cerebral blood flow, and deleterious effects on the permeability of the BBB. Phenyl-γ-valerolactone penetrates the BBB [342] and has functional attributes akin to other catechin catabolites that counter many of the earlier mentioned deleterious effects on brain tissues. Genomic and proteomic studies show that catechin metabolites have multimodal properties, modulating cellular pathways affecting cell adhesion, cytoskeletal organization, focal adhesion, endothelial permeability, and interaction with immune cells [225,226,343].

Flavonoids have therapeutic properties through their antioxidant and anti-inflammatory properties demonstrated in vitro. Some of the flavonoids can penetrate the blood–brain barrier from the systemic circulation to enter the brain directly, however, in general the bioavailability of intact flavonoids is limited due to poor absorption [29,344,345,346,347,348,349]. However, fortuitous generation of flavonoid metabolites by gut microbes that retain antioxidant and anti-inflammatory activity also needs to be considered in the overall therapeutic utility of these compounds [18,341,350,351,352,353,354]. Of the flavonoids, the isoflavones are the most bioavailable, however, anthocyanins and galloylated catechins are very poorly absorbed but can be converted into bioactive metabolites with therapeutic potential by gut microbes [345,346,347]. Gut microbes thus have important roles to play in the transformation and utilization of natural dietary flavonoids through the diverse enzyme systems that process these components in the gut [351,353,354]. Flavonoids generally cannot be metabolized effectively by human digestive enzyme systems but they can be transformed by enzymes produced by gut microorganisms into bioactive metabolites that can be transported by the gut-brain axis, or they can enter the systemic circulation from the gut and be transported to the brain where they more effectively penetrate the blood–brain barrier and thus have improved bioavailability and therapeutic utility [29,351,353,354]. Flavonoid metabolites that retain or exceed the antioxidant and anti-inflammatory capacity of the intact flavonoid indicate these have potent therapeutic potential. Furthermore, some flavonoid metabolites display biological activities not evident in the native flavonoid, which can be of therapeutic utility in the treatment of pathological neurodegenerative features in AD, PD, and HD by inhibiting the assembly and promoting the disassembly of protein aggregates in these disorders, reducing apoptosis of neurons and improving memory reducing cognitive decline in these neurodegenerative diseases [20,36,37,38,190,194,197,207,355,356,357].

## 14. Bioactive Flavonoid Metabolites and Regulation of Microbiome Bacterial Populations

The gut microbiome is a community of symbiotic microorganisms that inhabit the large intestine. These microbes have important roles to play in the maintenance of gut barrier integrity, inflammation, lipid and carbohydrate metabolism, immunity, and protection from pathogenic organisms. Colonization of the gut by pathogenic bacteria can lead to gut dysbiosis, significant alterations in gut bacterial populations, and an increase in the development of several diseases.

## 15. Metabolites Generated from Ellagic Acid with Bioactive Properties

Urolithin A has recently been approved as a functional food ingredient. Urolithin A and B have both been shown to improve metabolic functions and the maintenance of a healthy gut microbiome [358,359]. Uro-A and B also improve liver and kidney functions and induce the growth of *Akkermansia muciniphila*, a human mucin-degrading bacterium with health-promoting properties. Strategies have been developed to increase levels of *Akkermansia muciniphila* in the gut that counter obesity, diabetes, inflammation, and metabolic disorders [360,361,362]. A number of human and animal studies have shown that the abundance of *A. muciniphila* in the gut can be enhanced through dietary intervention. *A muciniphila* is available as a probiotic supplement [106].

*Gordonibacter urolithinfaciens* and *Ellagibacter isourolithinifaciens* are two human gut bacterial species that convert ellagic acid into urolithins [363].

## 16. Bioactive Quercetin Metabolites

Quercetin is a flavonoid that has been extensively examined in many studies that have demonstrated its antioxidant and anti-inflammatory properties in therapeutic applications. It is only relatively recently that interest has focused on quercetin metabolites and their biological properties [364]. The use of quercetin to treat OA rats has also been shown to influence gut bacterial populations with an elevation in the numbers of members of the *Clostridia, Bacteroidia*, and *Bacilli* families [365]. An increase in the number of quercetin metabolite species was also noted. A total of 94 human gut bacterial species have been examined for their ability to biotransform quercetin into different metabolites. *Bacillus glycinifermentans*, *Flavonifractor plautii*, *Bacteroides eggerthii*, *Olsenella scatoligenes,* and *Eubacterium eligens* were all shown to be capable of transforming quercetin into a number of metabolites displaying antiproliferative anticancer properties [366].

## 17. Flavonoids can Induce Neuroinflammation

While flavonoids have been shown to have many favourable properties in the alleviation of neuroinflammation and neurodegenerative processes [350,367,368], in a few cases gut bacteria have also been shown to detrimentally impact on inflammation in the brain, thus any prospective procedures conducted with flavonoids as therapeutics need to be carefully evaluated. In these cases of flavonoid-induced neuroinflammation, the balance of inflammatory cytokines in the gut and changes in intestinal and blood–brain barrier permeability can all produce detrimental impacts promoting the neuroinflammatory process. Having a healthy gut microbiome helps to prevent such detrimental effects on brain health [369,370,371].

## 18. Cellular Transport of Flavonoids by ATP-Binding Cassette (ABC)

### Transporter Proteins and their Potential Roles in the Modulation of Cellular Influx/Efflux in Disease Processes

A total of 30 lactic acid bacterial strains transform punicalagin in pomegranate extracts into ellagic acid and urolithins [372]. Proteomic analysis showed that this resulted in an increase in transglycosylases with potential hydrolytic roles in the target phenolic compound. An increase in levels of ATP-binding cassette (ABC) transporters was also observed and these may be relevant cellular transporters for flavonoid metabolites. Nine ABC transporter genes have been identified with proposed roles in the transport of flavonoid metabolites in *Salvia miltiorrhiza*. It is proposed that these control the distribution of pigmentation in this plant genus, and ABC transporter proteins may have similar roles to play in mammalian tissues [373]. An increase in ABC transporters has been observed in sows fed a diet rich in fermented Chinese medicine herbal additives, which also resulted in an increase in the antibacterial and anti-inflammatory properties of the milk these animals produced, improving milk quality [374]. ABC transporter proteins have proposed roles in flavonoid transport in the suppression of colitis and its transformation into colon cancer induced by kaempferol, increasing its bioavailability and efficacy [375]. *Streptococcus suis* (*S. suis*) is a highly virulent zoonotic pathogen that causes severe economic losses in the swine industry and is a public health concern with the rise of antibacterial antibiotic resistant strains that may transfer up the food chain to humans [376]. EGCG has antibacterial and other health benefits, significantly reducing the hemolytic activity of *S. suis,* and has been suggested as a potential treatment of *S. suis* infection. Laboratory investigations have shown ABC transporters have active roles to play in the mechanism of action of EGCG [376]. Kaempferol displays antibacterial activity against *H. pylori* with an action comparable to that of clarithromycin and amoxicillin. ATP-binding cassette transporters, flagellar assembly, and fatty acid metabolism are the major pathways in which *H. pylori* cells are responsive to kaempferol treatment. ABC transporters thus have key roles to play in the antibacterial action of kaempferol [377]. Inhibition of drug-efflux membrane transporters by prenylated flavonoids and their interactions with azole antifungals has been suggested as an approach to chemosensitize multidrug-resistant *C. albicans* strains that otherwise can be difficult to treat clinically [378]. The antibacterial basis of flavonoids has been shown to be due to their disruptive effects on bacterial efflux pumps [379]. Furthermore, the multi-drug resistance conferred by the P-glycoprotein efflux pump is a major cause of failure of cancer chemotherapy treatments. The multi-drug resistance-reversing activity of isobavachalcone through inhibition of the action of P-glycoprotein thus holds promise in the development of more effective anticancer treatment strategies using specific flavonoids [380].

## 19. Bioavailability of Flavonoids

Many native flavonoids display beneficial neuro-therapeutic effects and are capable of penetrating the BBB, however, concerns have been raised on the poor bioavailability of some flavonoid members. In general, polyphenolic compounds show a low bioavailability due to their interaction with other dietary components, and with phase I and II metabolic processes mediated by the liver, intestine, and microbiota. However, bioactive flavonoid metabolites generated by the gut microbiome that retain antioxidant and antiphlogistic properties of the native flavonoid occur, and these are of therapeutic value [381]. Methylation of flavonoid aglycones upon ingestion may improve their bioavailability to cells compared to the native glycosylated form [382]. Native anthocyanidins of low bioavailability may be converted to metabolites with improved bioavailability and are more easily absorbed by the gut epithelium microcapilleries and by the stomach, kidney and liver [383]. Citrus flavonoids, such as hesperidin, naringin and nobiletin, display a number of health benefits, including antioxidative, anti-inflammatory, and neuroprotective properties, however, they also have limited bioavailability with a large proportion of dietary flavonoids remaining unabsorbed in the colon [384,385]. Fortunately the gut microbiota convert these into bioactive fragments that are more readily absorbed.

Protocatechuic acid, a simple phenolic acid, is one such example of a bioactive flavonoid metabolite (Figure 10). Protocatechuic acid remains in the circulation for significantly longer periods and at higher concentrations than parent flavonoids, and it easily crosses the blood–brain barrier. Experimental studies strongly support the role of protocatechuic acid in the prevention of neurodegenerative processes affecting AD and PD [386,387]. Protocatechuic acid inhibits detrimental processes leading to cognitive and behavioral impairment, including accumulation of β-amyloid plaques in brain tissue, hyperphosphorylation of tau protein in neurons, excessive ROS generation, and neuroinflammation. Growing evidence shows protocatechuic acid is efficacious in the treatment of neurodegeneration and a safe substance with antineurodegenerative compound warrant further investigation [237]. Protocatechuic acid is a widely distributed, naturally occurring flavonoid metabolite active pharmacological component with antioxidant and anti-inflammatory properties. Protocatechuic acid can be generated from a number of flavonoids. Over the past two decades, there have been an increasing numbers of publications demonstrating the importance of flavonoids and their metabolites in biomedical applications [236]. Protocatechuic acid can be produced from many flavonoid metabolites in hibiscus but has not been specifically examined as a biotherapeutic from this tissue source. Phenolic hibiscus extracts possess inhibitory activities against acetylcholinesterase, butyrylcholinesterase, monoamine oxidase, and ecto-5′ nucleotidase memory-enhancing, antineuroinflammatory, antioxidative properties [388].

## 20. Conversion of Quercetin to Bioactive Metabolites by the Gut Microbiome

Quercetin is processed by gut microbiome members to a number of metabolites with improved bioavailability which retain the antioxidant properties of quercetin (Figure 10).

## 21. Processing of Epigallocatechin Gallate by the Gut Microbiome

As already discussed, the gut microbiome can generate EGCG metabolites that improve cerebrovascular function and have therapeutic value in the treatment of neurodegenerative disorders (Figure 11). Epicatechin improves cognitive functions, lowering the risk of AD or stroke, however, the biologically active molecular forms of epicatechin responsible are poorly understood [389]. γ-Valerolactone metabolites of EGCG modulate cellular regulation, cytoskeleton organization, focal adhesion, cell signaling, endothelial cell permeability, and interactions with immune cells [225]. Two major EGCG metabolites generated by the gut microbiome, 5-(4′-hydroxyphenyl)-γ-valerolactone-3′-sulfate and 5-(4′-hydroxyphenyl)-γ-valerolactone-3′-O-glucuronide [286] have high bioavailability and anti-inflammatory potency, decrease blood pressure and improve cerebrovascular blood flow [288]. Cerebrovascular dysfunction accelerates brain atrophy, reduces cognitive capability and increases risk of stroke, AD and dementia. EGCG in animal models [286,289] maintains neurocognitive function in aging rats, decreases risk of AD, and improves cerebrovascular blood flow [291,292,293]. Gut microbiome members with EGCG transforming properties include *Enterobacter aerogenes, Raoultella planticola, Klebsiella pneumoniae susp. pneumoniae*, and *Bifidobacterium longum subsp. Infantis* [311].

Table 2 summarises the major beneficial properties of plant flavonoids and gut microbiome generated flavonoid metabolites clearly showing their therapeutic potential. particularly in the alleviation of symptoms generated in neurological disorders of cognitive decline. Studies examining these compounds as biomedicines is warranted based on the data uncovered in this review. There is some urgency in undertaking these studies, with the ever increasing incidence of neurological disorders in the ageing global general population. A global burden of disease study showed the overall burden of global neurological disorders has increased significantly in the last decade from 1990 to 2019 with 10 million deaths and 349 million disability-adjusted life years due to neurological disorders reported [390], and very significant projected increases in the incidence of AD and dementia calculated to increase from 57.4 million cases globally in 2019 to 152.8 million cases in 2050 [391].

## 22. Conclusions

The complexity of phenolic and flavonoid compounds of therapeutic utility in the prevention of tissue degeneration or infection is huge. The problems of bioavailability limiting the therapeutic utility of flavonoids may be overcome by potent flavonoid metabolites that retain the antioxidant and anti-inflammatory potency of the native flavonoid. Furthermore, new biological activities displayed by the flavonoid metabolite not evident in the native flavonoid may extend the therapeutic utility of this class of compound. Studies are warranted to examine this aspect of gut microbiome-generated flavonoid metabolites and may be particularly useful in the treatment of neurological disorders. Therapeutic probiotics may be a means of engineering microbiome members that produce the beneficial flavonoid metabolites outlined in this review as a means of selectively treating neurological disorders. Thus mood, anxiety, and neurological disorders that result in cognitive deficits and motor dysfunction may potentially be targeted using such an approach. Therapeutic nutraceuticals that enhance the levels of these beneficial flavonoid metabolites may also be an approach worth investigation to improve overall health and wellbeing.

The importance of maintaining a dominant population of beneficial gut symbionts to prevent establishment of pathogenic organisms in the gut microbiome [423,424] has recently become apparent in a study which showed *Klebsiella aerogenes* producing 3β-hydroxysteroid dehydrogenase degraded estradiol leading to depression in menopausal female mice [425].

## Figures and Tables

**Figure 1 antioxidants-12-00663-f001:**
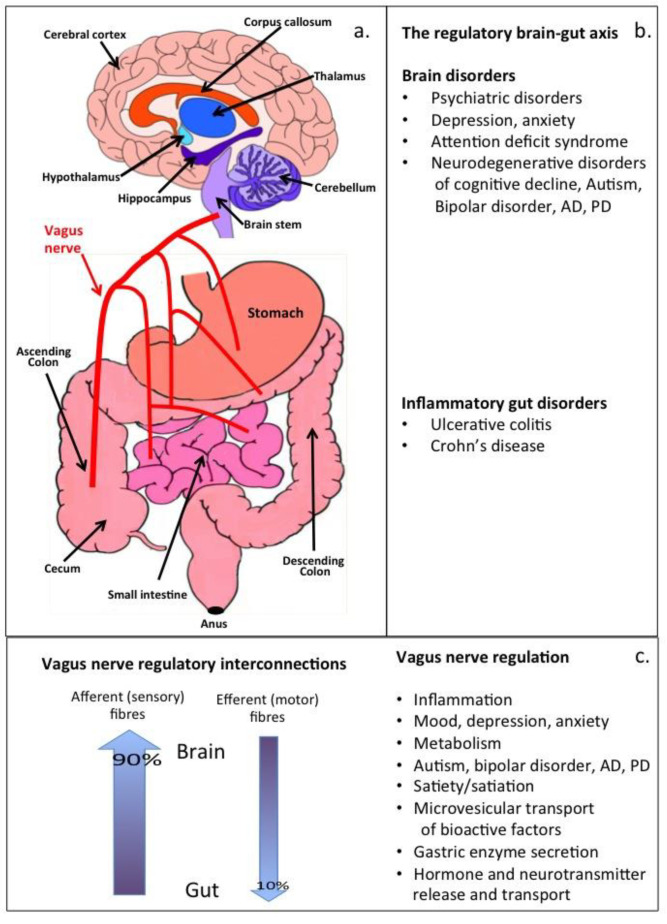
Schematic of the gut-brain axis: (**a**) demonstration of bidirectional communication by the parasympathetic vagus nerve, and some of the neurodegenerative conditions treated by vagal stimulation; (**b**) the vagal nerve transports compounds (generated from dietary flavonoids by the gut microbiome) of therapeutic value in the treatment of neurological disorders. Neurodegenerative conditions treated successfully by vagal stimulation are also shown (**c**).

**Figure 2 antioxidants-12-00663-f002:**
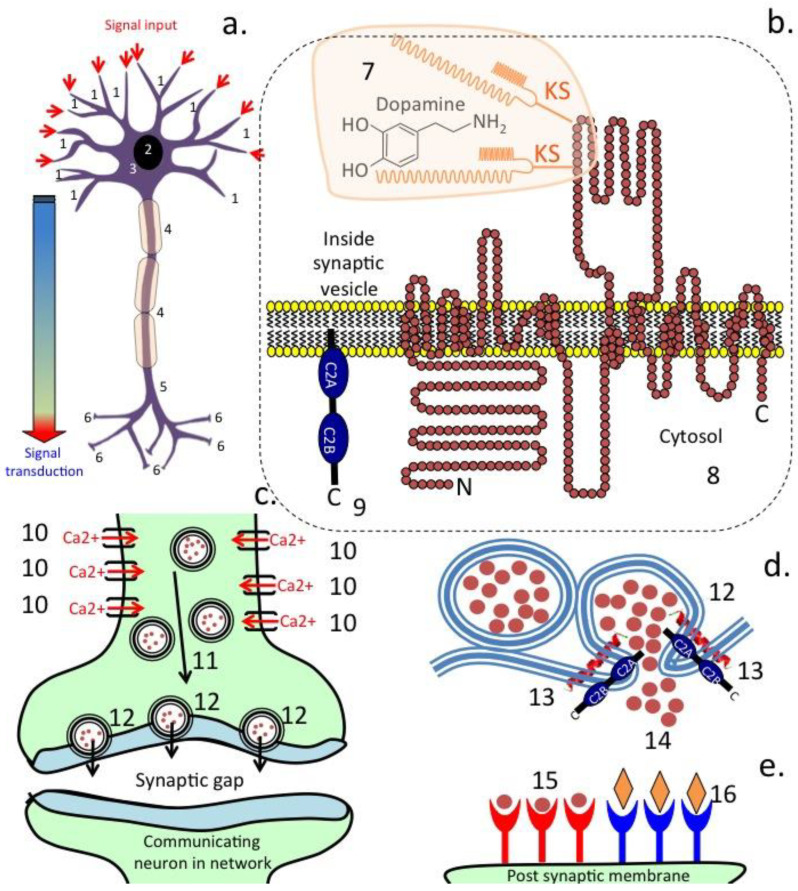
Schematic of neural signal transduction. Depiction of a neuron and its functional components (**a**) and the processes that occurs when a nerve is activated and signal transduction occurs (**b**–**e**). Specific features of the neuron are annotated, including the neural dendrite processes (1) where signal input occurs, the nucleus (2), which regulates neural activity in the neural cell body or soma (3). The myelinated sheath (4) covering the axon (5) ensures neural signal transmission efficiency is maintained. Neural synapses (6) communicate with other neurons in the neural network. Neural transmitters, such as dopamine (7), are stored in a smart gel matrix within the synaptic vesicle supplied by a 12 span transmembrane KS-storage and transport proteoglycan, SV-2 (8). The synaptic vesicle also has a calcium sensing glycoprotein: synaptotagmin (9). When a nerve becomes activated, the cell membrane becomes depolarized in the soma and a wave of membrane depolarization travels down the axon to the synapses. An influx of Ca^2+^ (10) into the nerve cytosol occurs in neuronal activation; this increase in Ca^2+^ is detected by synaptotagmin, which mobilises the transport of synaptic vesicles to the synaptic gap by SV-2 (11), and the synaptic vesicles fuse with the de-polarised pre-synaptic membrane (12). This fusion process is regulated by synaptotagmin and SNARE complex (SNAP Receptor) proteins and the neurotransmitters are released into the synaptic gap (14) to be taken up by neurotransmitter receptors on a communicating neuron in the network and the signal is successfully transduced. This is an extremely rapid process occurring in ~50–60 milliseconds.

**Figure 3 antioxidants-12-00663-f003:**
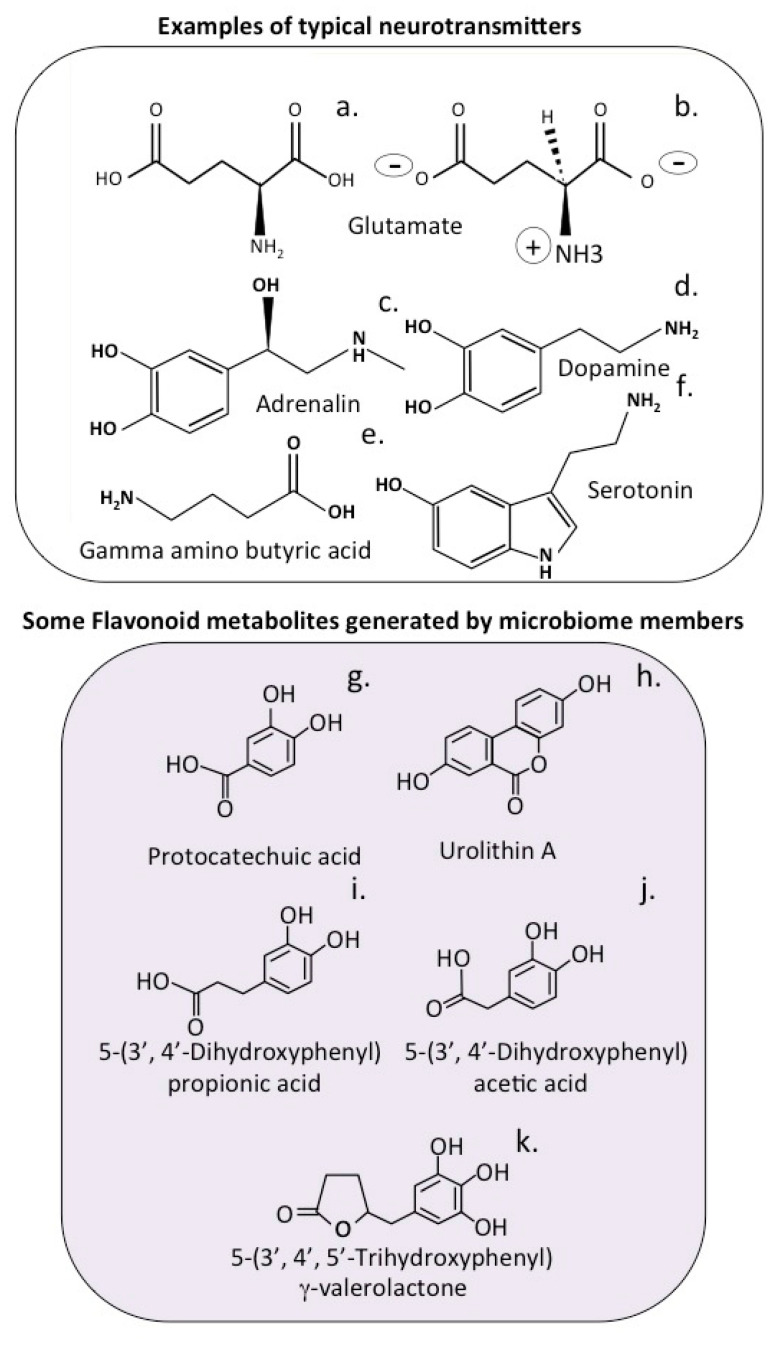
Neurotransmitters and flavonoid metabolites. A comparison of the structure of neurotransmitters (**a**–**f**) conveyed by nerves by vesicular transport, as shown in Figure 2. A few selected flavonoid metabolites generated by the gut microbiome are also shown for comparison (**g**–**k**). These flavonoid metabolites display a range of activities against neurons and cerebrovascular endothelial cells, and have beneficial properties that combat neuroinflammation, are neuroprotective, and have vasodilatory properties that promote cerebral blood flow in neurological disorders. Some of these metabolites have also been shown to promote mitochondrial biogenesis, improving neural bioenergetics and neuronal function in disorders, such as AD and PD, where a cognitive decline has been observed.

**Figure 4 antioxidants-12-00663-f004:**
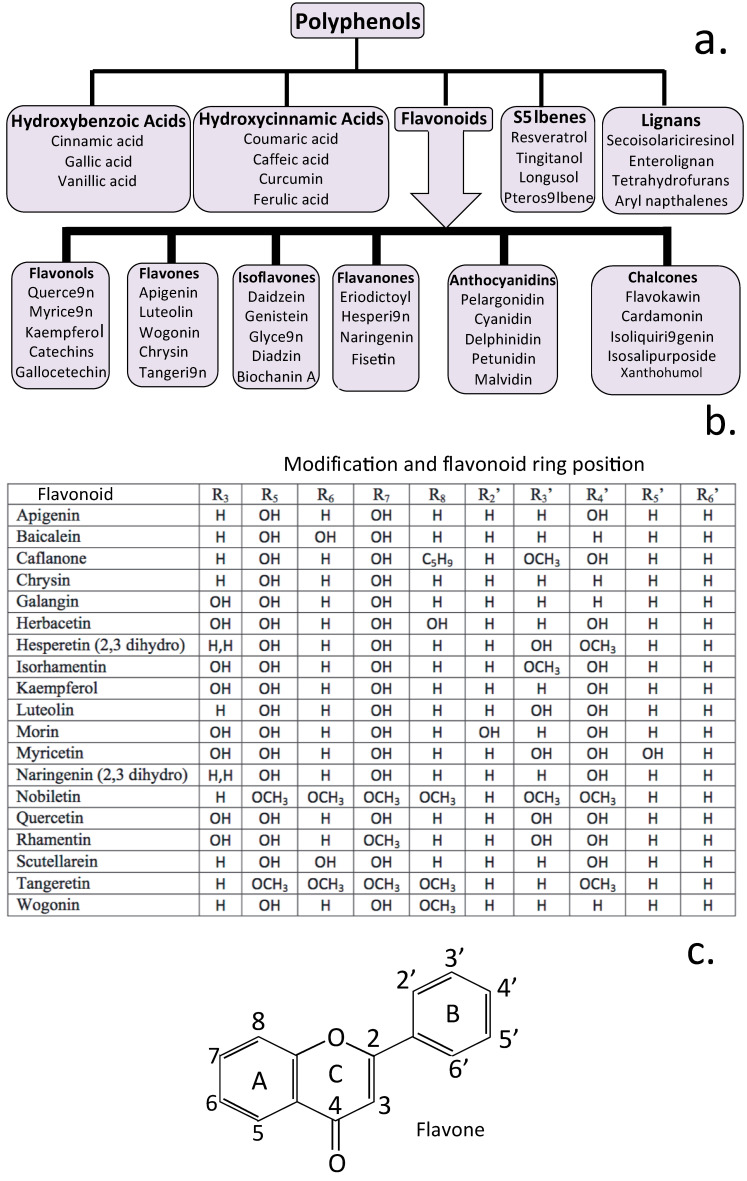
The flavonoids. Classification of the flavonoids, a major sub-category of phenolic compounds (**a**), showing the diverse modifications (**b**) that occur on the A, B and C flavone ring structures (**c**).

**Figure 5 antioxidants-12-00663-f005:**
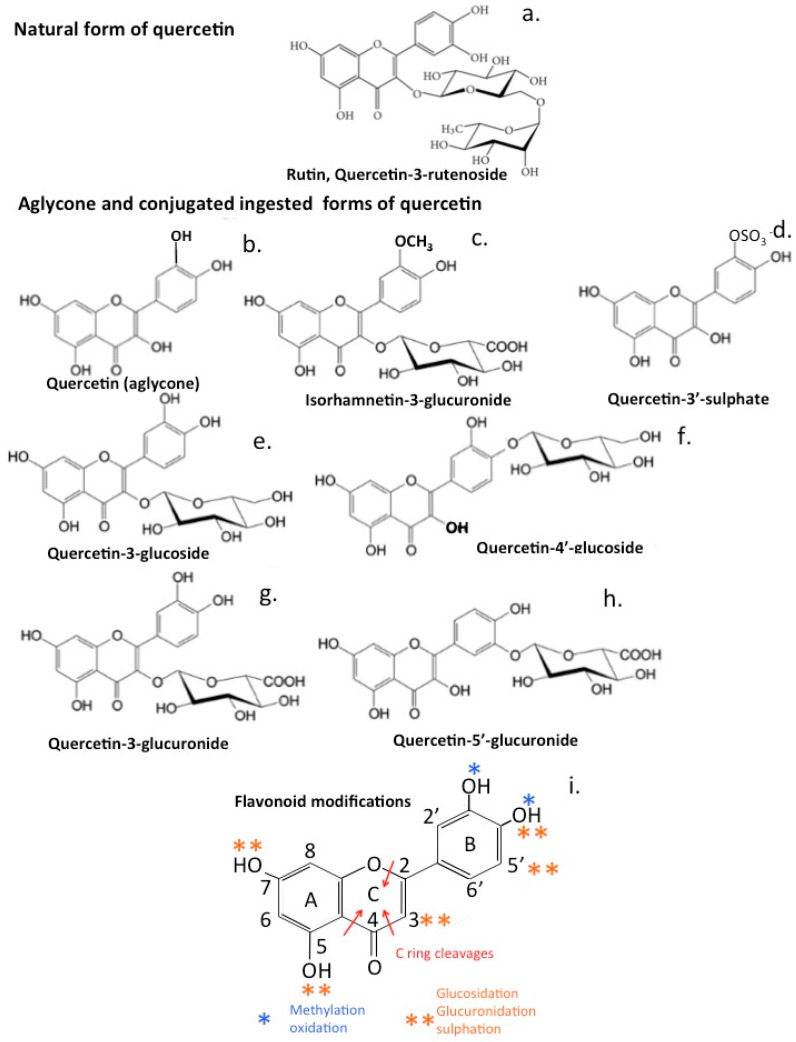
The many forms of quercetin. As a representative flavonoid, quercetin occurs as a glycosylated form (rutin) in plant tissues (**a**), which, when ingested, is converted to the aglycone form (**b**), isorhamnetin-3-glucuronide (**c**), quercetin also undergoes sulfation (**d**), or glucosidation (**e**,**f**), and glucuronidation (**g**,**h**), as shown at specific locations in the flavone A, B, and C rings (**i**). The C-ring may undergo cleavages at the positions shown when the flavone is processed by the gut microbiome.

**Figure 6 antioxidants-12-00663-f006:**
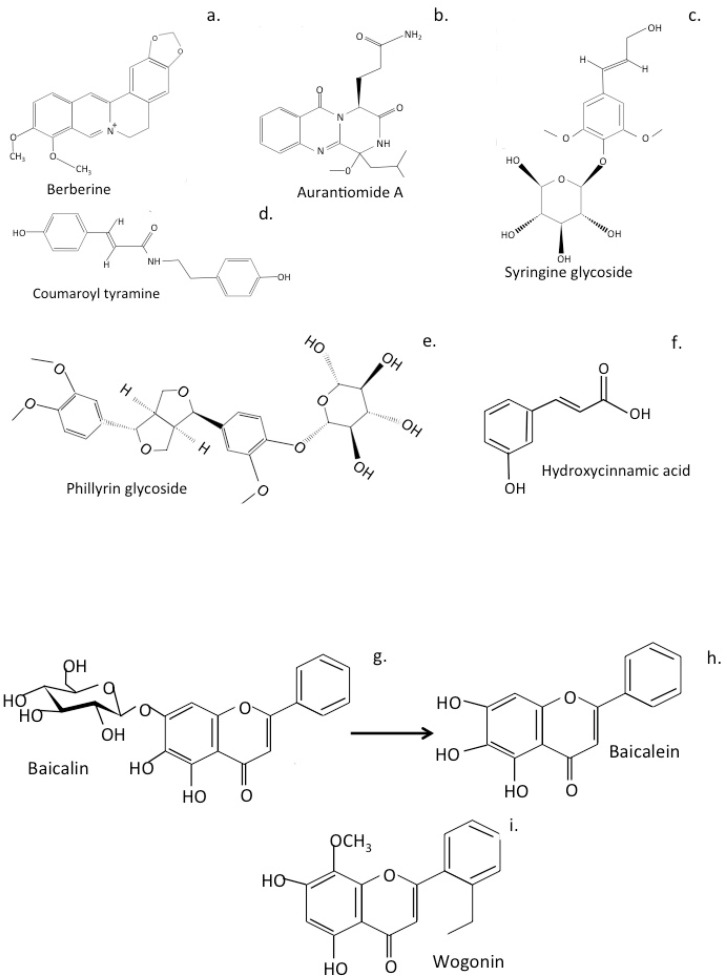
Examples of some of the pharmacologically active complex polyphenolic compounds that have been identified in LeZhe Chinese complimentary medicine herbal preparations (**a**–**f**) and Shuang-Huang-Lian herbal preparations (**g**–**i**) used to treat neurodegenerative conditions and respiratory infections.

**Figure 7 antioxidants-12-00663-f007:**
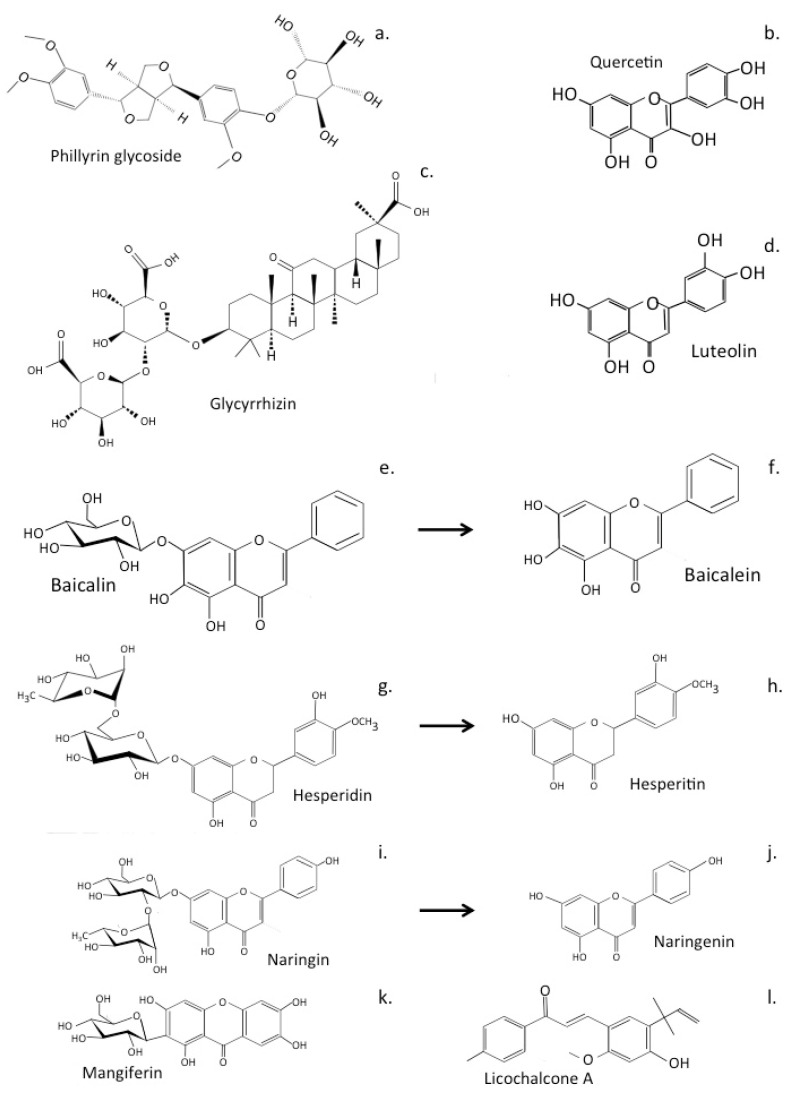
Examples of the varied ring structures of some of the flavonoids covered in this review. Structures of bioactive phenolic compounds identified in Chaihu-Shugan-San, Qingfei Paidu, and Ma Xing Shi Gan traditional Chinese complimentary medical formulations. Identification of phillyrin glycoside, quercetin, glycyrrhizic acid, luteolin, baicalin, baicalein, hesperidin, hesperitin, naringin, naringenin mangiferin, and licochalcone A (**a**–**l**) as bioactive components of such herbal formulations.

**Figure 8 antioxidants-12-00663-f008:**
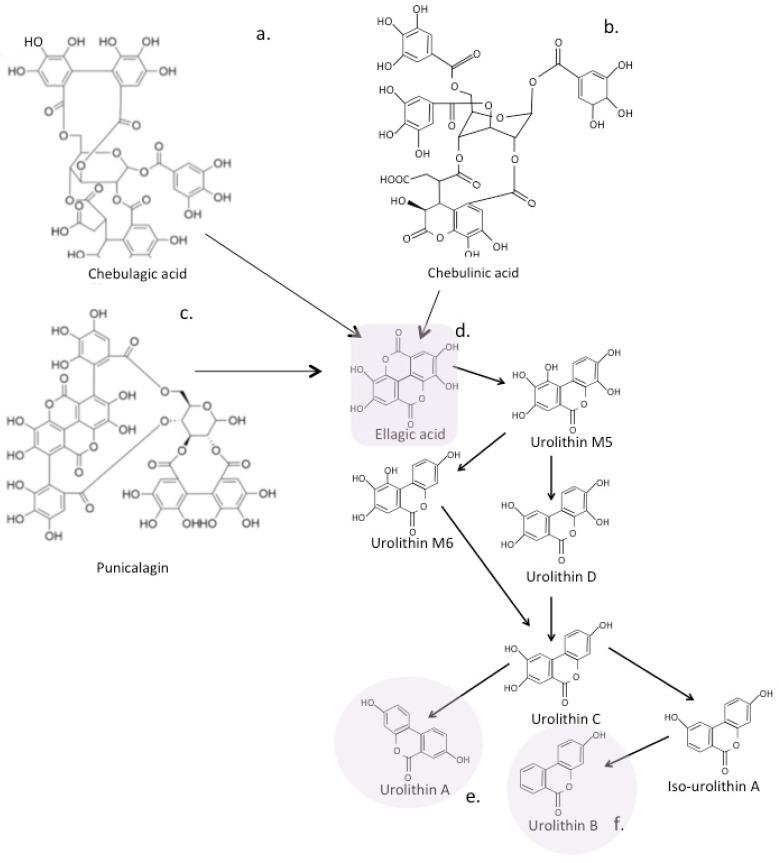
Generation of Urolithins: complex heterocyclic elligatannin compounds, such as chebulagic acid (**a**), chebulinic acid (**b**), and punicalagin (**c**) occurring in plant foods are converted to ellagic acid (**d**) and a number of urolithin metabolites by the gut microbiome. Urolithin A (**e**) and B (**f**) are active in neural tissues.

**Figure 9 antioxidants-12-00663-f009:**
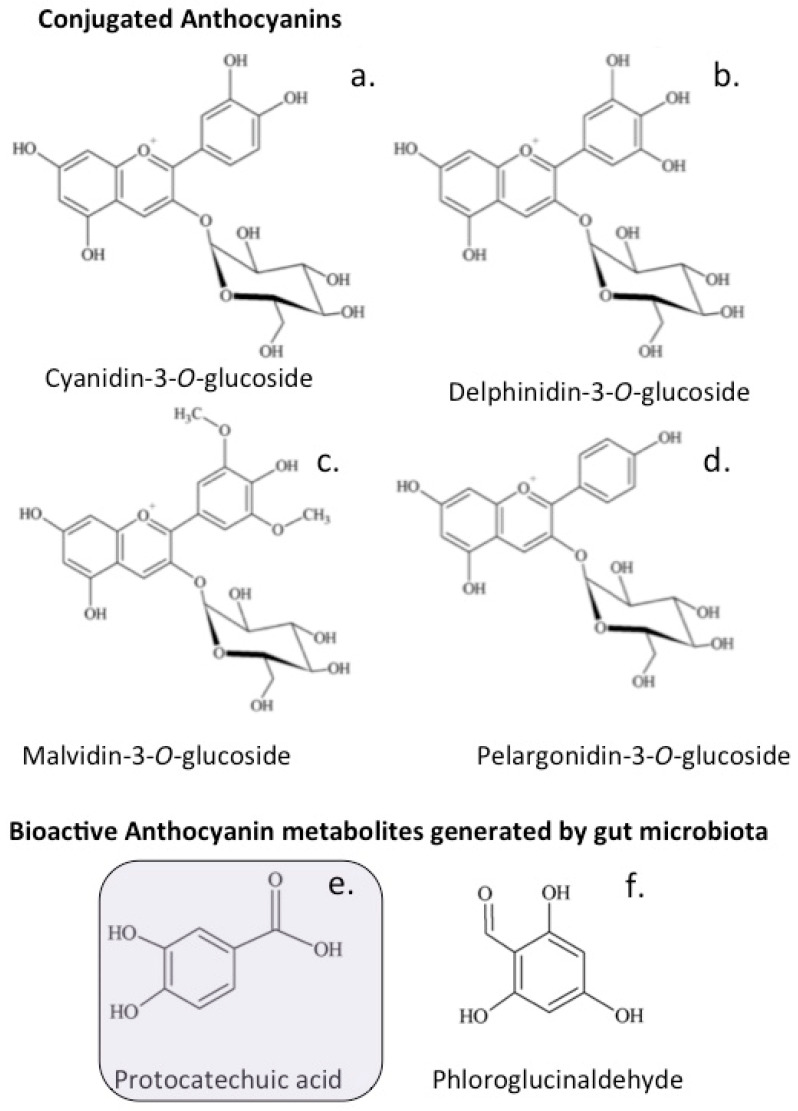
Multicyclic structural forms of the anthocyanins and their bioactive protocatechuic acid and phlorglucinaldehyde metabolites generated by the gut microbiota. Cyanidin-3-*O*-glucosied (**a**); Delphinidin-3-*O*-glucosied (**b**); Malvidin-3-*O*-glucosied (**c**); Pelargonidin-3-*O*-glucosied (**d**); protocatechuic acid (**e**); Phloroglucinaldehyde (**f**).

**Figure 10 antioxidants-12-00663-f010:**
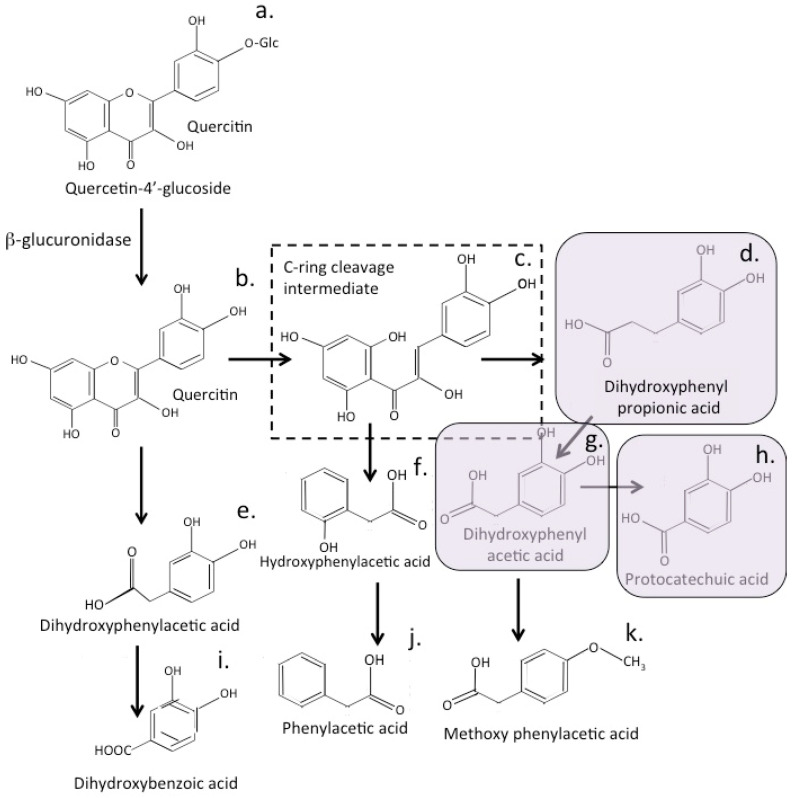
Degradative pathways used by *C. perfringens* and *B. fragilis* of the human gut microbiome to process quercetin into bioactive metabolites, as proposed by Peng et al. [325]. Quercetin-4’glucuronide (**a**) is initially degraded by β-glucuronidase to form quercetin aglycone (**b**) then dihydroxyphenyl acetic acid (**e**) and dihydroxybenzoic acid (**i**). Alternatatively C ring internal cleavage of quercetin aglycone into an intermediate form (**c**) can also be converted to hydroxyphenyl acetic acid (**f**) and phenyl acetic acid (**j**) or to hydroxyphenyl propionic acid (**d**) then dihydroxyphenyl acetic acid (**g**) then to methoxy phenylacetic acid (**k**) or protocatechuic acid (**h**).

**Figure 11 antioxidants-12-00663-f011:**
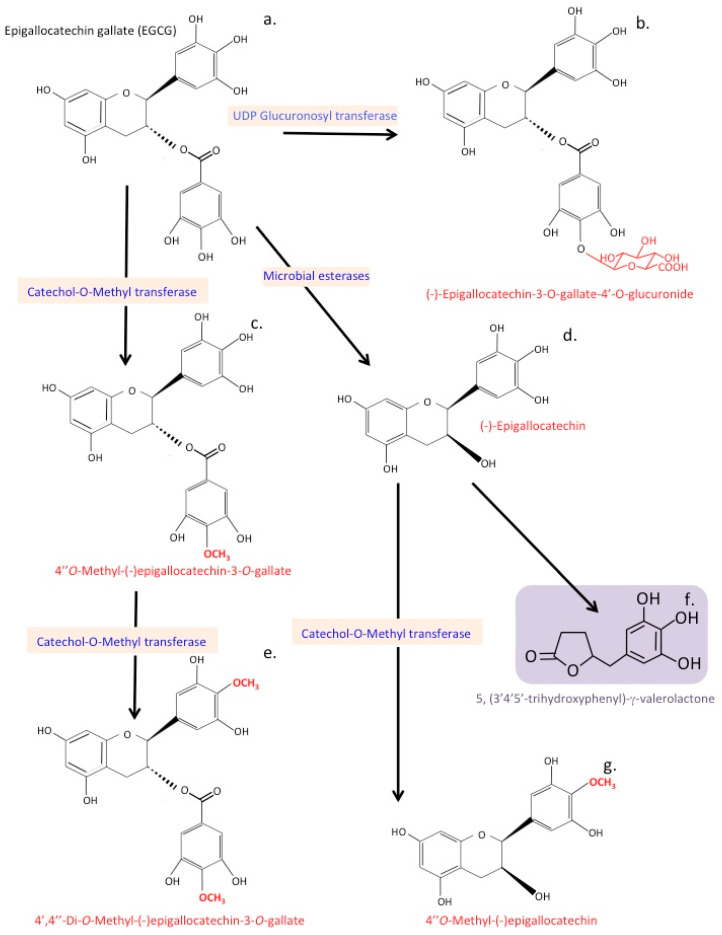
Generation of tea EGCG metabolites by the gut microbiome, as proposed by Lotito et al. 2011 [227] 5(3,4,5-trihydroxyphenyl)-g-valerolactone has therapeutic pharmacological properties useful in cancer therapy, antioxidant free radical scavenging, and cerebrovascular therapeutic applications in neurodegenerative disorders. (**a**) Epigallocatechin gallate (EGCG); (**b**) (-)-Epigallocatechin-3-O-gallate-4’-*O*-glucuroide; (**c**) 4’’*O*-Methyl-(-)epigallocatechin-3-*O*-gallate; (**d**) (-)-Epigallocatechin; (**e**) 4’,4’’-Di-*O*-Methyl-(-)epigallocatechin-3-*O*-gallate; (**f**) 5, (3’4’5’-trihydroxyphenyl)-γ-vcalerolactone; (**g**) 4’’*O*-Methyl-(-)epigallocatechin.

**Table 1 antioxidants-12-00663-t001:** Flavonoids that induce Nrf2 expression.

Compound	Flavonoid Class	Reference
Quercetin	Flavonol	[119,120,121]
Myrcetin		[122]
Kaempferol		[123,124]
Catechin		[125]
Gallocatechin		[126]
Apigenin	Flavone	[127]
Luteolin		[128,129,130]
Wogonin		[131,132]
Chrysin		[133,134]
Baicalin		[135]
Diadzein	Isoflavone	[135]
Genistein		[136,137,138]
Biochanin A		[139]
Hesperidin	Flavonone	[139,140]
Hesperitin		[141,142,143]
Naringenin		[144,145]
Pelargonidin	Anthocyanidin	[146]
Cyanidin		[147]
Delphinidin		[148]
Petunidin		[149]
Cardamonin	Chalcone	[150,151]
Xanthohumol		[152,153]
Isoliquiritigenin		[154]

**Table 2 antioxidants-12-00663-t002:** A summation of the gut microbiome and processing of therapeutic dietary components which generates bioactive flavonoid metabolites that promote neuronal health and counter neurological deficits.

Neurological Disorder	How the Gut Microbiome Impacts the Disorder	Ref.
AD	Induction of Nrf2 expression by flavonoids is neuroprotective countering neuroinflammation. Flavonoids also have intrinsic antioxidant activity against generation of ROS by COX, LOX, MPO, XO. Many microbiome generated flavonoid metabolites retain or have enhanced or new bioactive properties not evident in the native flavonoid and greater bioavailability. Protocatechuic acid is present in the circulation at higher concentrations for significantly longer than native flavonoids and easily crosses the BBB, inhibits accumulation of β-amyloid plaques, hyperphosphorylation of tau protein in neurons and excessive generation of neuroinflammatory ROS, has potent antioxidant and anti-inflammatory properties, is neuroprotective, increases neuronal proliferation and inhibits apoptosis of neural stem cells. Urolithin A potently inhibits the pro-oxidant heme peroxidases MPO and LPO reducing tissue inflammation and significantly reduces phorbol myristate acetate stimulated ROS generated by neutrophils. Urolithin B is a MAO inhibitor and improves cognitive deficits. Urolithin M5 is a neuraminidase inhibitor, urolithin M6 is an inhibitor of LDH. Urolithins are neuroprotective, inhibit Aβ_25-35_-induced neurotoxicity and neurodegenerative MAO activity. Urolithin A promotes mitophagy and mitochondrial biogenic neuronal function. γ-valerolactones detoxify the effects of amyloid β oligomers. Some flavonoid metabolites have vasodilatory properties that improve cerebrovascular circulation and lower blood pressure.	[20,53,392,393,394,395,396,397,398,399]
PD	Anti-oxidant properties of flavonoids and metabolites and ability to induce Nrf2 expression is neuroprotective. Gut microbiome generated components may potentially regulate α-synuclein folding lowering the levels of misfolded α-synuclein deposition in pathological protein aggregates leading to neurotoxicity and a decline in neural function. Induction of mitochondrial biogenesis by flavonoid species promotes neuronal bioenergetics and viability.	[54,400,401,402]
Autism	Modulation of the gut microbiota to deliver high-fat low-carbohydrate ketogenic products has proven beneficial in countering the deficits in communication and social interaction evident in autism.	[403,404,405,406]
Bipolar disorder	Diets rich in n-3 fatty acids, folate, S-adenosylmethionine, N-acetyl cysteine and probiotic mediated effects offer promising interventions in the treatment of bipolar disorder	[84,407,408,409,410]
Depression and Anxiety	Symptoms of depression and anxiety have been shown to be linked to alterations in the microbiota and can be treated by probiotic dietary manipulation with certain food products known as psychobiotics.	[390,391,411,412,413]
Epilepsy	Manipulation of the gut microbiome to deliver a ketogenic high-fat low-carbohydrate diet mimics the fasting state of the body and is beneficial in treatment of drug-resistant epilepsy.	[55,414,415,416]
Stroke	Anti-oxidant, anti-thrombotic and vasodilatory properties of flavones and flavone metabolites may lower possibility of stroke and improve vascular repair processes. The anti-oxidant, anti-inflammatory, neuroprotective properties of quercetin may minimize the incidence of ischemic stroke. Promotion of endothelial cells by flavonoids improves vascular repair processes.	[417,418,419,420,421,422]

Abbreviations used: Nrf2, nuclear factor erythroid 2–related factor 2; ROS, reactive oxygen species; COX, cyclooxygenase; LOX, lipoxygenases; MPO, myeloperoxidase; XO, xanthine oxidase; LPO, lactoperoxidase; BBB, blood brain barrier; MAO, myeloperoxidase’ LDH, lactate dehydrogenase.

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
