# Peer review of "The Potential of Flavonoids and Flavonoid Metabolites in the Treatment of Neurodegenerative Pathology in Disorders of Cognitive Decline"

_antioxidants, 2023, doi:10.3390/antiox12030663_

Round 1
Reviewer 1 Report
Despite a very interesting topic I cannot recommend this review for publication at least in its present form because of the following reasons.
1. The title does not correspond to the content of this review, which is not focused on the announced topic. The review scratches on the surface of the problem and does not highlight the role of microbiota generated flavonoids metabolites as a new therapeutic frontier in the treatment of neurodegenerative diseases.
2. Structurally, it is not well organized and mechanisms of bidirectional communications between the gut and the brain are poorly analyzed. In the context of the announced topic some of statements are really confusing. See for example, lines 66-67: “Bioactive compounds are transported to the brain by efferent vagal fibres to stimulate specific brain regions”. Traditional routes of microbiota metabolites including their intestinal absorbance and subsequent appearance in circulation have not been considered at all. In this context I address the author to the legend to Fig. 1 from the review by Bonaz et al. (2018) (ref. 5 in this review).
3. The “starring” of n. vagus in this review is not clear for me because a role of flavonoids and their targets have not been adequately considered.
4. Many important statements are not supported by appropriate citations. For example, see Lines 535-536: “Several reports indicate flavonoids improve cognitive functions, inhibit or delay the formation of pathological amyloid beta aggregates or neurofibrillary tangles improving neural function”. However, no references are given to support this information, while other (less important) statements are supported by too many references, see below: Lines 99-100: “the existence of a gut-liver and gut-brain regulatory connection that exerts some measure of control over linked organ systems has received considerable attention [24-42]”.
5. “A new therapeutic frontier in the treatment of neurodegenerative disorders” is based on results of studies unrelated to “flavonoid metabolites delivered by the gut-brain axis”. For example, Section 5.6 describes effects of a plant flavonoid Plumbagin. However, I did not find any indications on its linkage to the gut microbiome.
In my opinion, the author should consider first microbiota generated flavonoid metabolites and routes by which these flavonoid metabolites reach the brain. Ironically, a reasonably good plan of the review is given in the abstract (lines 12-28), but the author did not follow it.
Author Response
Despite a very interesting topic I cannot recommend this review for publication at least in its present form because of the following reasons.
Reviewer comment
The title does not correspond to the content of this review, which is not focused on the announced topic. The review scratches on the surface of the problem and does not highlight the role of microbiota generated flavonoids metabolites as a new therapeutic frontier in the treatment of neurodegenerative diseases.
Author response
Reviewer comment
Structurally, it is not well organized and mechanisms of bidirectional communications between the gut and the brain are poorly analyzed. In the context of the announced topic some of statements are really confusing. See for example, lines 66-67: “Bioactive compounds are transported to the brain by efferent vagal fibres to stimulate specific brain regions”. Traditional routes of microbiota metabolites including their intestinal absorbance and subsequent appearance in circulation have not been considered at all. In this context I address the author to the legend to Fig. 1 from the review by Bonaz et al. (2018) (ref. 5 in this review).
Author response
I was aware of the absorption route from the systemic circulation however I have added a revised section on the bioavailability of flavonoids to the revised manuscript-this is highlighted.
Reviewer comment
The “starring” of n. vagus in this review is not clear for me because a role of flavonoids and their targets have not been adequately considered.
Author response
I have added comments on alternative routes of administration of flavonoids and their metabolites other than the vagus nerve.
Reviewer comment
Many important statements are not supported by appropriate citations. For example, see Lines 535-536: “Several reports indicate flavonoids improve cognitive functions, inhibit or delay the formation of pathological amyloid beta aggregates or neurofibrillary tangles improving neural function”. However, no references are given to support this information, while other (less important) statements are supported by too many references, see below: Lines 99-100: “the existence of a gut-liver and gut-brain regulatory connection that exerts some measure of control over linked organ systems has received considerable attention [24-42]”.
Author response
I have addressed your comments by re-organising citations to specific regions in the revised manuscript in a more appropriate more balanced manner as you correctly pointed out.
Reviewer comment
“A new therapeutic frontier in the treatment of neurodegenerative disorders” is based on results of studies unrelated to “flavonoid metabolites delivered by the gut-brain axis”. For example, Section 5.6 describes effects of a plant flavonoid Plumbagin. However, I did not find any indications on its linkage to the gut microbiome.
Author response
We have removed comments on plumbagin from the revised manuscript since this is used therapeutically directly as a plant infusion. Comments on plumbagin were originally in the manuscript since this is a small plant compound of similar size and structure to flavonoid metabolites discussed in this review and it has useful biological activity and has been used in traditional medical applications demonstrating that such small metabolites retain important biological properties.
Reviewer comment
In my opinion, the author should consider first microbiota generated flavonoid metabolites and routes by which these flavonoid metabolites reach the brain. Ironically, a reasonably good plan of the review is given in the abstract (lines 12-28), but the author did not follow it.
Author response
I have added a new segment to the revised manuscript to reflect routes of administration of flavonoid metabolites to the brain.
Reviewer 2 Report
In this research, Melrose J. concluded that microbiome-generated flavonoid metabolites (e.g. proto-catechuic acid, urolithins, g-valerolactones) retain the anti-oxidant and anti-inflammatory potency of the native flavonoid and with additional bioactive properties could also promote mitochondrial health and cerebrovascular microcapillary function. This manuscript has an agreeable study design and logical hypothesis. However, the main question remains: how could it be scientifically approved that microbiome-mediated flavonoid metabolites serve a protective effect in individuals? Has any human or animal study supported this protective effect directly via the production of flavonoid metabolites? Moreover, answering the below question could improve the quality of this research.
Major:
1. Why was the author in the manuscript using the general word “gut microbiota”, and no name of the specific bacteria was mentioned? For example, which bacteria from gut microbiota can convert ETs to ellagic acid?
2. Why did you separate some parts of the manuscript into Chinese and western medicine? For example, is gut microbiota in Chinese different from western people?
3. From my point of we, you start to say something related to your topic from page 14 of this paper, where you described polyphenolic precursors compound that is processed by the gut microbiota. Why are you again focused on chemical compounds innated of bacteria that convert them to specific metabolites?
4. Why is Part 7.1 and figure 10 only placed in this manuscript related to the title?
5. Why you don’t mention any bacteria in all metabolite pathways, including figure 11?
6. Adding a column in table 2 and introducing related bacteria among gut microbiota could add some vital information.
7. Whilst the exact neuroprotective mechanisms of some flavonoid metabolites are fully understood, the gut microbiota is also known to mediate neuroinflammation, and the production of these metabolites is not guaranteed protective effects in disease conditions. So why did the authors not report any data regarding adverse or side effects of gut microbiota-related flavonoid metabolites in this study?
Author Response
Antioxidants Reviewer 2 Comments
Reviewer comment
The main question remains: how could it be scientifically approved that microbiome-mediated flavonoid metabolites serve a protective effect in individuals? Has any human or animal study supported this protective effect directly via the production of flavonoid metabolites? Moreover, answering the below question could improve the quality of this research.
Author response
The anti-oxidant and anti-inflammatory properties of flavonoids and their metabolites have been determined from in-vitro studies, preclinical studies using animal models and human clinical trials and the importance of the gut microbiome in the generation of flavonoid metabolites is firmly established. Flavonoid metabolites are also beginning to be evaluated for their therapeutic properties with promising findings so far. The biological activity of specific flavonoid metabolites has also been determined from in-vitro studies, a few studies have also examined the efficacy of specific flavonoid metabolites as therapeutic agents in animal models. An extra segment covering this material has been added to the revised manuscript- this is highlighted.
Reviewer comment
Major:
Why was the author in the manuscript using the general word “gut microbiota”, and no name of the specific bacteria was mentioned? For example, which bacteria from gut microbiota can convert ETs to ellagic acid?
Author response
If the reviewer looks at the legends to figures we did actually identify species that undertook the proposed pathways indicated in the figure. However we have expanded on this identifying specific gut bacteria in relevant segments of the revised manuscript.
Reviewer comment
Why did you separate some parts of the manuscript into Chinese and western medicine? For example, is gut microbiota in Chinese different from western people?
Author response
The Chinese traditional medicinal preparations were treated separately from flavonoids that have been examined in Western medicine since Traditional Chinese medicinal preparations are very complex mixtures of pharmacologically active compounds whereas Western medical studies adopt a more minimalistic approach looking at individual compounds to better understand their therapeutic properties and molecular targets. In contrast, traditional medical preparations may contain up to 10 different herbal components with each of these having many pharmacological agents (possibly up to 20). Thus the Chinese herbal medications are extremely complex and quite different to medications that have been examined in Western medicine. This is why I made comments in the manuscript outlining how network pharmacology is being utilised to identify individual bioactive components and how these may interact with one another and several hundred molecular targets in such traditional Chinese medicines. I was not inferring that the microbiome in the Chinese population differed from that found in Western populations.
Reviewer comment
From my point of we, you start to say something related to your topic from page 14 of this paper, where you described polyphenolic precursors compound that is processed by the gut microbiota. Why are you again focused on chemical compounds innated of bacteria that convert them to specific metabolites?
Author response
Flavonoids are members of the polyphenolic family of molecules numbering in excess of 8000 compounds, examination of individual flavonoids and their metabolites more precisely identifies therapeutic compounds. It was beyond the scope of this review to examine all polyphenolic compounds.
Reviewer comment
Why is Part 7.1 and figure 10 only placed in this manuscript related to the title?
Author response
The revised manuscript is now re-organised.
Reviewer comment
Why you don’t mention any bacteria in all metabolite pathways, including figure 11?
Author response
Comments on bacterial species has been added to the revised manuscript.
Reviewer comment
Adding a column in table 2 and introducing related bacteria among gut microbiota could add some vital information.
Author response
In the studies cited in this table they generally did not all comment on the gut microbiome members responsible for the observations that were made. I therefore decided on against adding an additional column in this table covering such material however have made comments on specific microbiome members of interest and particular flavonoids elsewhere in the revision to address this comment.
Reviewer comment
Whilst the exact neuroprotective mechanisms of some flavonoid metabolites are fully understood, the gut microbiota is also known to mediate neuroinflammation, and the production of these metabolites is not guaranteed protective effects in disease conditions. So why did the authors not report any data regarding adverse or side effects of gut microbiota-related flavonoid metabolites in this study?
Author response
Comments to address this have been added to the revised manuscript.
Round 2
Reviewer 1 Report
In my opinion the author made some cosmetic changes that insignificantly improved the manuscript.
I believe that the data summarized in this review may be reorganized and have to be reorganized provided that the Title is really reflects what the author wants to consider. Based on the information available in the abstract, I suggest the following plan:
(1) Flavonoids are a biodiverse family of dietary compounds that have anti-oxidant, anti-inflammatory, anti-viral and anti-bacterial cell protective profiles.
(2) Flavonoids are potential therapeutic agents in biomedicine and have been widely used in traditional complimentary medicine for generations.
(3) The gut microbiome plays an important role in the generation of bioactive flavonoid metabolites retaining or exceeding the anti-oxidative and anti-inflammatory properties of the intact flavonoid and in some cases new anti-tumor and anti-neurodegenerative bioactivities.
(4) Certain food items have been identified with high pre-biotic profiles suggesting that neutraceutical supplementation may be beneficially employed to preserve a healthy population of bacterial symbiont species and minimize establishment of harmful pathogenic organisms.
(5) Gut health is an important consideration effecting overall health and well-being of linked organ systems.
Other comments: The suggestion about possible involvement of neurotransmitter transporting systems in transport of flavonoid metabolites generated by the gut microbiota is highly speculative and must be justified by citation of corresponding scientific reports demonstrating such unique/unusual possibility.
Reviewer 2 Report
The following points should still be addressed:
Minor concerns:
1. The title is much too long and should be shortened.
2. Part two of the title '' The Treatment of Neurodegenerative Pathology In Disorders of Cognitive Decline'' still has insufficient information and evidence in the manuscript. The illustrated results make less sense to pick out individual target genes/proteins related to specific flavonoids, as these necessarily reflect the overall picture of the possible treatments. This aspect should be included and discussed in the appropriate paragraph.
Round 3
Reviewer 1 Report
Although I still do not understand why the Introduction section begins with nervus vagus instead of highlighting the role of flavonoids and their gut microbiome generated metabolites, the author has made some improvements to the paper, which may be accepted for publication. However, there are typing errors (e.g. see Fig. 3), which need corrections.